# Discontinuities in hygroscopic growth below and above water saturation for laboratory surrogates of oligomers in organic atmospheric aerosols

5 Natasha Hodas,[1,2] Andreas Zuend,[3] Katherine Schilling,[1,a] Thomas Berkemeier,[4] Manabu Shiraiwa,[4,5] Richard C. Flagan,[1,6] John H. Seinfeld[1,6]

[1]Division of Chemistry and Chemical Engineering, California Institute of Technology, Pasadena, CA, USA
[2]Department of Environmental Sciences and Management, Portland State University, Portland, OR, USA
10 [3]Department of Atmospheric and Oceanic Sciences, McGill University, Montreal, Quebec, Canada
[4]Multiphase Chemistry Department, Max Planck Institute for Chemistry, Mainz, Germany
[5]Department of Chemistry, University of California Irvine, CA, USA
[6]Division of Engineering and Applied Science, California Institute of Technology, Pasadena, CA, USA
[a]now at: United States Army Criminal Investigation Laboratory, Forest Park, GA, USA

*Correspondence to:* Natasha Hodas (nhodas@pdx.edu)

**Abstract.** Discontinuities in apparent hygroscopicity below and above water saturation have been observed for organic and mixed organic-inorganic aerosol particles in both laboratory studies and in the ambient atmosphere. However, uncertainty remains regarding the factors that contribute to observations of low 20 hygroscopic growth below water saturation, but enhanced cloud condensation nuclei (CCN) activity for a given aerosol population. Utilizing laboratory surrogates for oligomers in atmospheric aerosols, we explore the extent to which such discontinuities are influenced by organic component molecular mass and viscosity, non-ideal thermodynamic interactions between aerosol components, and the combination of these factors. Measurements of hygroscopic growth under subsaturated conditions and the CCN activity of aerosols 25 comprised of polyethylene glycol (PEG) with average molecular masses ranging from 200 to 10,000 g mol$^{-1}$ and mixtures of PEG with ammonium sulfate (AS) were conducted. Experimental results are compared to calculations of hygroscopic growth at thermodynamic equilibrium conducted with the Aerosol Inorganic Organic Mixtures Functional groups Activity Coefficients (AIOMFAC) model, and the potential influence 30 of kinetic limitations to observed water uptake was further explored through estimations of water diffusivity in the PEG oligomers. Particle phase behavior, including the prevalence of liquid-liquid phase separation (LLPS), was also modeled with AIOMFAC. Under subsaturated RH conditions, we observed little variability in hygroscopic growth across PEG systems with different molecular masses; however, an increase in CCN activity with increasing PEG molecular mass was observed. This effect is most 35 pronounced for PEG-AS mixtures, and, in fact, an enhancement in CCN activity was observed for the PEG10000-AS mixture as compared to pure AS, as evidenced by a 15% reduction in critical activation diameter at a supersaturation of 0.8%. We also observed a marked increase in apparent hygroscopicity for mixtures of higher molecular mass PEG and AS under supersaturated conditions as compared to subsaturated hygroscopic growth. AIOMFAC-based predictions and estimations of water diffusivity in 40 PEG suggest that such discontinuities in apparent hygroscopicity above and below water saturation can be attributed, at least in part, to differences in the sensitivity of water uptake behavior to surface tension

effects. There is no evidence that kinetic limitations to water uptake due to the presence of viscous aerosol components influenced hygroscopic growth. For the systems that display an enhancement in apparent hygroscopicity above water saturation, LLPS is predicted to persist to high RH. This indicates a miscibility gap and is likely to influence bulk-to-surface partitioning of PEG at high RH, impacting droplet surface tension and CCN activity. This work provides insight into the factors likely to be contributing to discontinuities in aerosol water-uptake behavior below and above water saturation that have been observed previously in the ambient atmosphere.

**1 Introduction**

The extent to which interactions between airborne aerosols and water vapor modulate the Earth's radiation budget is a source of uncertainty in projections of the impact of aerosols on radiative forcing (Boucher et al., 2013). The uptake of water in relative humidity (RH) regimes below water saturation (RH < 100%) affects aerosol particle size distributions and optical properties, impacting the efficiency of scattering and absorption of solar radiation. Under supersaturated RH conditions relevant to the activation of cloud condensation nuclei (CCN), aerosol properties influence cloud droplet number, cloud albedo, and, potentially, cloud lifetime. Further, condensed-phase water present in atmospheric aerosols and cloud droplets serves as a medium into which reactive organic gases can partition and undergo aqueous-phase chemistry to form secondary organic aerosol (SOA) (McNeill, 2015 and references therein). While the water-uptake behavior of inorganic aerosol components is generally well characterized (Seinfeld and Pandis, 2016), a more thorough understanding of the influence of organic compounds on aerosol hygroscopicity and CCN activity is needed.

A complicating factor in the understanding and representation of the water-uptake behavior of organic and mixed organic-inorganic aerosols is the fact that such particles can exist in a variety of phase states. Non-ideal thermodynamic interactions between organic and inorganic particle components can result in liquid-liquid phase separation (LLPS) in which inorganic-dominated and organic-dominated phases coexist (Erdakos and Pankow, 2004; Ciobanu et al., 2009; Zuend et al., 2010; Bertram et al., 2011; Pöhlker et al., 2012; Song et al., 2012; Zuend and Seinfeld, 2012; You et al., 2012, 2013, 2014). Moreover, organic aerosol components can exists as viscous liquids, semisolids, and glasses depending on their composition and ambient conditions (e.g., temperature and RH) (Zobrist et al., 2008, 2011; Mikhailov et al., 2009; Virtanen et al., 2010; Koop et al., 2011; Tong et al., 2011; Saukko et al., 2012; Song et al., 2015; Zhang et al., 2015).

Variability in the phase states of atmospheric aerosols is expected to influence their hygroscopicity. For example, inhibition of mass transfer through viscous liquids or semisolid particles may result in kinetic limitations to the uptake and evaporation of water (Koop et al., 2011; Tong et al., 2011;

Bones et al., 2012; Krieger, et al., 2012; Pöschl and Shiraiwa, 2015). As a result, the timescales and mechanisms of condensation and evaporation may be different for liquid and solid or semisolid particles (Shiraiwa et al., 2013). Previous studies, for example, have observed extended time scales for equilibration with water vapor and/or kinetic limitations to the crystallization of ammonium sulfate for particles

containing sucrose (glass transition temperature [$T_g$] = 331 – 335.7 K; Zobrist et al., 2008; Dette et al., 2014) (Tong et al., 2011; Bones et al., 2012; Robinson et al., 2014; Hodas et al., 2015).

Variability in water vapor uptake with particle physical state also influences the activation and growth of CCN and cloud droplets (Bilde and Svenningsson, 2004; Berkemeier et al., 2014) and, thus, may impact the microphysical properties of clouds by modulating droplet number concentration. Kinetic

limitations to water uptake (e.g., due to slowed rates of droplet growth) have been shown to result in as much as a 30% increase in CCN activation dry particle diameter and a decrease in cloud droplet growth rates by a factor of two (Nenes et al., 2001; Asa-Awuku et al., 2009; Raatikainen et al., 2012). Not accounting for non-ideal interactions between particle components (i.e., assuming equilibrium partitioning to an ideal solution under circumstances in which this assumption is not valid) can result in a 10 - 40% over

prediction of cloud droplet number, depending on aerosol loading (Nenes et al., 2001). On the other hand, the presence of surface-active organic components can contribute to enhancements in CCN activity by reducing the surface tension of the particle surface-air interface (Ma et al., 2013; Sareen et al., 2013; Woo et al., 2013). In addition, simultaneous condensation of semivolatile organic vapors and water onto aerosol particles may enhance water uptake by increasing the availability of soluble material (Topping and

McFiggans, 2012; Topping et al., 2013). This effect is expected to increase with increasing RH because the atmospheric conditions leading to higher RH (e.g., decreasing temperature) also lead to decreases in organic compound vapor pressures and, therefore, the condensation of increasingly volatile material.

Measurements of the hygroscopic growth of atmospheric aerosols in both sub- and supersaturated conditions have demonstrated discontinuities in water-uptake behavior below and above water saturation

(Good et al., 2010; Irwin et al., 2010, 2011; Dusek et al., 2011; Ovadnevaite et al., 2011; Hersey et al., 2013). Specifically, for a given population of aerosols, previous studies have observed low degrees of hygroscopic growth below water saturation, but high CCN activity. Hersey et al., (2013) measured sub- and supersaturated hygroscopicity in an airborne campaign over the Los Angeles basin and observed reductions in subsaturated hygroscopic growth with increasing photochemical age of SOA and for biomass-burning

aerosol, but increases in aerosol CCN activity under these same circumstances. Similarly, Good et al. (2010) found that the use of the single parameter, κ (Petters and Kreidenweis, 2007), to describe both sub- and supersaturated hygroscopic growth, as is common in models of aerosol-cloud interactions, was not sufficient to capture the water-uptake behavior of marine aerosols.

Several explanations have been put forth to reconcile observed discontinuities in water uptake

below and above water saturation. A recent study suggested that for some semisolid particles (as characterized by bounce fraction), the mechanism of water uptake differs under conditions above and below water saturation, with adsorption dominating under subsaturated conditions (at RH < 95%) and

absorption dominating under conditions relevant to CCN activation (Pajunoja et al., 2015). In that work, slightly oxygenated SOA derived from α-pinene and longifolene displayed water-uptake behavior under subsaturated conditions similar to that of particles comprised of $SiO_2$, which are known to take up water by surface adsorption. Frenkel-Halsey-Hill adsorption theory was able to describe subsaturated hygroscopic

growth for these particles (Pajunoja et al., 2015). Another factor potentially contributing to discontinuities in water-uptake behavior is that water uptake, as modeled with the Köhler equation, is sensitive to different parameters at low (<95%) and high (≥95%) RH, with the effects of surface tension being negligible at low RH, but important determinants of CCN activity (Wex et al., 2008). Recent work suggests that a compressed film model that accounts for the presence of surface tension lowering organic compounds at

the air-droplet interface during CCN activation is able to reconcile previously observed differences in apparent hygroscopicity derived from CCN measurements and subsaturated hygroscopic growth measurements (Ruehl et al., 2016). Similarly, non-ideal thermodynamic interactions are expected to be of greater importance under the more concentrated conditions relevant to subsaturated hygroscopic growth as compared to supersaturated conditions (Wex et al., 2008; Petters et al., 2009a). It has been hypothesized

that differences in water-uptake behavior above and below water saturation arise from variability in the prevalence of LLPS and/or the presence of solid or semisolid aerosol components with RH and temperature. For example, Renbaum-Wolff et al. (2016) observed that such discontinuities are pronounced for aerosol systems that underwent LLPS at high RH. Others have suggested that high particle viscosity at subsaturated RH values can inhibit water uptake, but this effect is reduced as particle viscosity decreases

with increased RH and particle liquid water content (Virtanen et al., 2010; Koop et al., 2011), possibly explaining the lower hygroscopic growth at subsaturated RH values as compared to supersaturated growth previously observed. Variability in water-uptake kinetics with RH has important implications for the activation and growth of CCN, as it suggests that the hygroscopic behavior of some particles can shift as ambient conditions transition from a subsaturated to a supersaturated regime (e.g., in an ascending air

parcel).

High molecular mass compounds, such as organic oligomers, are a potential source of both viscous and surface-active atmospheric aerosol components. Oligomers with molecular masses ranging from 200 to 1600 g/mol have been detected in SOA generated in laboratory studies from a variety of precursors, with these compounds constituting between 25 and 70% of SOA mass (Hallquist et al., 2009

and references therein). SOA components with properties indicative of oligomers, as well as the mixture of high molecular mass compounds (likely including oligomers) termed humic-like substances (HULIS) have also been observed in the atmosphere (Kroll and Seinfeld, 2008 and references therein; Hallquist et al., 2009 and references therein; Lee et al., 2015). Barsanti and Pankow (2004, 2005, 2006) suggested that particle-phase accretion reactions could explain the presence of oligomers and esters in SOA. Laboratory

studies also suggest aqueous-phase reactions lead to the formation of oligomers and esters in SOA (Altieri et al., 2008; Tan et al., 2010), with the formation of higher molecular mass compounds being favored under

the more concentrated conditions relevant to aerosol liquid water as compared to the more dilute conditions of cloud droplets (Tan et al., 2010).

Because molecular mass impacts volatility, solubility, and viscosity, a more thorough understanding of the properties of aerosol components with high molar masses is needed to accurately represent their behavior in large-scale atmospheric models. Utilizing laboratory surrogates for oligomers in atmospheric aerosols, we explore the influence of organic-component molecular mass and viscosity, LLPS, and the combined effects of these factors on hygroscopic growth and CCN activity and discuss the extent to which such factors were likely to be contributors to previously observed differences in water-uptake behavior below and above water saturation. Polyethylene glycol (PEG) was chosen as a model compound for this work because the availability of PEG with a range of polymer chain lengths/molecular masses allows for the comparison of water-uptake behavior across aerosol systems with differing viscosities, but otherwise similar chemical properties. Further, previous studies of PEG-AS particles (using optical and Raman microscopy) have shown that, at some RH values, such systems undergo LLPS in which a PEG shell fully engulfs an AS core (Ciobanu et al., 2009, 2010). Finally, PEG oligomers have been shown to have pure component surface tensions substantially lower than that of water (Wu, 1974; Winterhalter et al., 1995; Rey and May, 2010; Wu et al, 2011) and, thus, may impact water-uptake behavior by lowering the surface tension of the droplet-air interface.

**2 Methods**

**2.1 Aerosol systems**

To systematically study the influence of organic-component molecular mass and viscosity, LLPS, and the combined effects of these factors on water uptake under sub- and supersaturated RH conditions, the hygroscopic growth and CCN activity of particles comprised of PEG and mixtures of PEG and ammonium sulfate (AS) were measured with the Differential Aerosol Sizing and Hygroscopicity Spectrometer Probe (DASH-SP) and a Droplet Measurement Technologies Cloud Condensation Nuclei (DMT CCN) Counter, respectively. Experiments were conducted with aerosol systems containing PEG with average molecular masses of 200 ("PEG200"), 1,000 ("PEG1000"), and 10,000 ("PEG10000") g mol$^{-1}$. This corresponds to a range in $T_g$ from 208.15 K to 313.65 K (Pielichowski and Flejtuch, 2002; Dow, 2011). Under dry conditions and at room temperature, PEG200 is a liquid, PEG1000 is a waxy semisolid, and PEG10000 exists as solid flakes. In addition to the PEG systems, hygroscopic diameter growth factors (HGFs) and CCN activity were also measured for AS in control experiments to ensure proper instrument operation. All solutions used for aerosol generation were prepared by dissolving the reagents in Milli-Q water with resistivity ≥ 18.2 MΩ cm. For the mixed PEG-AS systems, the mass ratios of PEG:AS were 2:1 for all PEG molecular masses. The PEG oligomers were purchased from Sigma Aldrich and AS was purchased from Macron Fine Chemicals. It should be noted that the PEG are comprised of a mixture of polymers with a

range of molecular masses (190 – 210 g mol$^{-1}$ for PEG200, 950 – 1,050 g mol$^{-1}$ for PEG1000, and 8,500 – 11,500 g mol$^{-1}$ for PEG10000), with the number included in the name indicating the average molecular mass of the mixture. Aerosols were generated by atomizing the aqueous solutions. Previous work has suggested that incomplete mixing of aerosol components in aqueous solution and/or fractionation of

components during atomization can contribute to variability and uncertainty in hygroscopicity measurements for aerosol systems containing surface-active components, particularly for components with low water solubility (Petters and Petters, 2016). However, because PEG is highly water soluble, it is not expected that this was a significant contributor to uncertainty in experimental results. Before entering the DASH-SP or DMT CCN counter, atomized droplets passed through a silica gel diffusion dryer with a

residence time of approximately 3 to 5 s. HGF and CCN activity measurements were conducted at room temperature (~298 K).

## 2.2 Hygroscopic growth factor measurements

HGFs were measured with the DASH-SP (Sorooshian et al., 2008) at RH values ranging from 30 to 90% in increments of 10%. After entering the DASH-SP inlet, the aerosols are further dried in a Nafion dryer (with a residence time of 1 s), they pass through a $^{210}$Po neutralizer, and are then size-selected with a long-column differential mobility analyzer (DMA) based on their electrical mobility. For the HGF measurements described here, particles with dry mobility diameters of 250 nm were selected with the DMA. After size

selection, the monodisperse aerosol population is split into four humidified channels, one of which is kept dry. Particle size after exposure to elevated RHs in the humidified channels is then measured at each channel outlet with an optical particle counter (OPC). A minimum of 1500 particles is sized to generate the humidified size distributions. This was repeated 30 times within each experiment (i.e., for each aerosol system) at each RH value. OPC signal height, which is a function of both particle size and refractive index,

is inverted to give particle diameter using an empirical calibration surface relating OPC signal height, refractive index, and particle size (Sorooshian et al., 2008). This surface is generated in dry calibration experiments in which OPC signal heights for salts with known refractive indices are recorded for particles with diameters ranging from 200 to 500 nm. The calculation of wet particle diameter requires knowledge of the particle's dry effective refractive index. This is measured in the dry DASH-SP channel. Wet particle

diameter is determined from OPC pulse height and the calculated effective refractive index for the dry particle components by iterating on the 3-dimensional surface until agreement is achieved, within experimental uncertainty, between the wet effective refractive index as determined using this surface and a volume-weighted refractive index for the calculated wet size (taking into account the individual refractive indices for dry components and water) (Sorooshian et al, 2008). It is assumed that the particles are spherical

and that they scatter, but do not absorb light. The uncertainty in DASH-SP-derived droplet diameters has been shown to be ~8% at RH < 80% and ~5% at RH ≥ 80% (Sorooshian et al., 2008). HGFs were

calculated by dividing the wet particle diameter derived from DASH-SP measurements by the dry particle diameter (250 nm).

HGF measurements were carried out across 4 experiments with RH in the wet channels set to 30% and 40% ("experiment 1"), 50% and 60% ("experiment 2"), 70% and 80% ("experiment 3"), and 90% ("experiment 4"). For experiment 2, an overestimation in growth for the ammonium sulfate control (HGF > 1) led to the finding that average OPC pulse heights for the dry channel during those runs were 22% to 27% lower than average pulse heights obtained during the other experimental runs. For experiments 1, 3, and 4, on the other hand, dry-channel pulse heights for ammonium sulfate deviated by less than 5% across each experiment. The lower dry-channel pulse heights resulted in an underestimation of dry refractive index and, consequently, an overestimation of droplet growth. Similarly, dry-channel pulse heights were 12% – 14% lower on average for several of the PEG systems during experiment 2, while deviations across other experiments were typically ≤ 9%. Note that regardless of the humidified channel settings, it is expected that for a given aerosol system, the OPC pulse heights measured in the dry DASH-SP channel will be approximately the same across experiments. As a result, the values obtained during experiment 2 can be replaced by values obtained during other experimental runs without a substantial loss of information. Thus, for circumstances in which dry-channel pulse heights deviated by more than 10% from those measured during other experimental runs, the anomalous pulse heights were replaced by the average of the OPC pulse heights obtained during all other experimental runs; that is, the directly measured dry-channel pulse heights from experiment 2 were replaced by the value obtained by averaging pulse heights measured during experiments 1, 3, and 4. While average pulse heights are expected to produce a reasonable estimate of HGF, a higher degree of uncertainty in the HGFs existed when measured at RHs of 50% and 60%. As a result, we focus to a greater degree on the measurements conducted at the five other RH conditions.

**2.3 CCN Measurements**

The CCN activity of the aqueous PEG and PEG-AS aerosol systems was measured with a DMT CCN Counter (Roberts and Nenes, 2005). Similar to the DASH-SP, particles in the CCN counter are exposed to elevated RHs (in this case above water saturation, i.e., RH > 100%) and then counted and sized with an OPC. With the goal of characterizing water-uptake behavior across the full range of RH conditions occurring in the atmosphere, CCN measurements were conducted at a supersaturation of 0.8%, toward the upper-bound of supersaturations representative of typical cloud conditions (~0.1% – 1%). In the CCN counter, the supersaturation experienced by the particles is a function of both the temperature difference across the top and bottom of the instrument column and the flow rate within the column. All experiments were carried out at a total flow rate of 0.5 L min$^{-1}$, with a sheath flow:sample flow ratio of 10:1. Supersaturation calibrations using AS were conducted to determine the across-column temperature difference needed to achieve a supersaturation of 0.8%. For each aerosol system, CCN activity was characterized based on the critical dry activation diameter ($D_{crit}$), the diameter at which 50% of particles are

activated to form CCN at a constant supersaturation (i.e., here at 0.8%). Prior to entering the inlet of the CCN counter, particles were size-selected with a long-column DMA and CCN counts were obtained for particles with dry diameters between 20 and 210 nm with a spacing of 10 nm. All particles were assumed to be spherical. Due to the particle drying step prior to size selection in the DMA, the

crystallization of AS, as well as the potential for the higher molecular mass PEG oligomers to be present as solids, could result in a small degree of uncertainty in the actual sphere-equivalent diameter of particles transmitted through the DMA. Uncertainty in particle diameter is expected to be minimal, however, as shape factors for AS have been estimated to be between 1.0 (i.e., spherical) and 1.04 (Gysel et al., 2002; Zelenyuk et al., 2006). Shape factors for PEG-containing

submicron particles are unknown at present. A condensation particle counter (CPC) sampled particles in parallel with the CCN counter to provide total particle counts, and activation fractions were calculated from the ratio of CCN concentration ($C_{CCN}$) to total particle concentration measured with the CPC ($C_{CPC}$). Correction factors were applied to measured CCN concentrations to account for particle losses in the CCN counter based on the results of Brechtel and Kreidenweis (2000), who provided a polynomial curve

describing particle losses as a function of particle size. Loss-corrected measured activation fractions as a function of particle diameter were fit with a 5-parameter sigmoidal curve with the following form:

$$C_{CCN}/C_{CPC} = \text{MIN}\left\{ \frac{c_4 \times \sqrt{D_0} + c_3}{\left[1 + \left(\frac{D_0}{c_1}\right)^{c_2}\right]^{c_5}}, 1.0 \right\} \tag{1}$$

where $c_1$, $c_2$, $c_3$, $c_4$, and $c_5$ are fit parameters (Table A1) and $D_o$ is the dry particle diameter. $D_{crit}$, defined as the dry diameter associated with a 50% activation fraction, was determined using the best-fit curves. The

sigmoidal curve fits describe the experimental data well for all aerosol systems (Pearson r > 0.99 in all cases).

**2.4 Thermodynamic Modeling**

To explore the extent to which observed water uptake under subsaturated RH conditions can be explained by equilibrium thermodynamic partitioning, measured HGFs were compared to calculations of hygroscopic growth at thermodynamic equilibrium. Predictions of HGF by a thermodynamic equilibrium model for the systems studied – if sufficiently accurate – allow for an interpretation of the experimental results with

regard to particle viscosity. If the presence of viscosity-enhancing aerosol components (here PEG oligomers) sufficiently inhibited the uptake or evaporation of water, it is expected that measured HGFs would deviate substantially from those modeled at equilibrium. This model-measurement disagreement would occur if equilibrium growth was not achieved in the 4 s residence time of the DASH-SP humidifiers and/or if all water present in the atomized droplet could not be evaporated in the ~ 5 s total residence time

in the diffusion and Nafion dryers prior to sizing in the DMA. Thus, comparison of measured HGFs with HGFs predicted at thermodynamic equilibrium provides insight into the influence of aerosol viscosity on

water uptake behavior. The equilibrium model used in this study is based on the equilibrium gas-particle partitioning framework introduced by Zuend et al. (2010) and Zuend and Seinfeld (2012). The central component of this framework is the Aerosol Inorganic-Organic Mixtures Functional groups Activity Coefficients (AIOMFAC) model (Zuend et al., 2008, 2011).

Three types of chemical equilibria are accounted for by the model in the present study: (i) vapor-liquid equilibrium (VLE) of water, (ii) a potential liquid-liquid equilibrium (LLE) affecting all components in the condensed phase, and (iii) a potential solid-liquid equilibrium of ammonium sulfate. The gas phase is treated as an ideal mixture, and water vapor is the only gas-phase component for which equilibration with the particle phase is considered. The vapor pressures of the different PEG oligomers and of AS in liquid solution are sufficiently low at 298 K that evaporation on the timescale of the experiments (total aerosol residence time ~ 10 s) is negligible. VLE of an aqueous bulk solution with a gas phase implies equivalence of the mole-fraction based water activity, $a_w$, with the (controlled) RH in the gas phase.

The non-ideality of liquid phases is characterized by the AIOMFAC group-contribution model (i.e., organic molecules are represented as assemblies of functional groups) in which non-ideal interactions between inorganic ions and organic functional groups in an aqueous solution are taken into account through the calculation of activity coefficients. Activity coefficients of dissolved ions and solvent compounds are quantified based on the contributions of long-range, mid-range, and short-range molecular interactions to the Gibbs excess energy of a system (Zuend et al., 2008; 2011). Included in our modeling framework is the computation of a potential LLPS, as well as the co-existence of ammonium sulfate in the crystalline state in equilibrium with the liquid organic-inorganic phases. The existence or absence of a liquid-liquid phase equilibrium is determined by solving a system of nonlinear equations to determine the phase state (i.e. a single liquid phase, or two liquid phases at LLE) that achieves a minimization of the system's overall Gibbs energy (Zuend and Seinfeld, 2013). To summarize the liquid phase treatment, AIOMFAC is applied in the equilibrium model for the computation of activity coefficients of the different mixture species in one or two co-existing liquid phases at given overall PEG/AS mass ratios and RH values.

It is well-known that the water activity and phase equilibria of PEG oligomers and polymers in aqueous solutions are rather poorly described (e.g., Ninni et al., 1999) when the standard set of functional groups is used in the UNIversal quasi-chemical Functional group Activity Coefficient model (UNIFAC; Fredenslund et al., 1975; Hansen et al., 1991) and hence also in AIOMFAC, which includes a modified UNIFAC model. In order to provide an improved model representation of aqueous PEGs (of various polymer chain lengths) and of the ternary water + PEG + AS phase diagrams, a special oxyethylene group (-CH$_2$-O-CH$_2$- ; the repetitive monomer unit in PEG) was introduced in a recently developed PEG-specific AIOMFAC parameterization. Other system-specific AIOMFAC modifications were implemented and adjustable model parameters were determined based on published experimental data on water activities and LLE data of different water + PEG + AS systems at room temperature. A detailed description and discussion of this specific AIOMFAC parameterization will be given elsewhere (Zuend et al., *in preparation*).

Diameter HGFs, particle phase states, and phase compositions were calculated at RHs ranging from ~0 to 99% for all aerosol systems, with the exception of PEG10000-AS. For numerical and theoretical reasons, the current equilibrium model does not support a reliable calculation of LLE for high molar mass PEG oligomers. Instead, HGF calculations for the PEG10000-AS system were performed based on a Zdanovskii-Stokes-Robinson (ZSR)-like assumption, in which complete separation between phases comprised of PEG and AS is assumed at all RH values, and water uptake by the individual aerosol phases is treated separately. Total water uptake is then calculated assuming additivity of the water present in each phase. This simplification is not expected to contribute substantially to error in calculated HGFs, nor to impact model-measurement comparisons, as experimental LLE data for bulk PEG10000-AS systems suggest near-complete separation between AS and PEG up to high mole fractions of water that would be expected to occur at RH values outside the range studied in DASH-SP experiments (RH > ~95%) (Graber et al., 2007; Fig. A1).

Model predictions were conducted for both hydration (low to high RH) and dehydration conditions (high to low RH). For the AS-containing systems, the hydration-case calculations account for the existence of a crystalline AS phase at SLE with the liquid phase prior to complete deliquescence of AS. In contrast, the calculations referring to the dehydration branch in the experiments were performed in a mode that allows for supersaturated conditions with respect to SLE of AS up to a critical supersaturation beyond which crystallization (efflorescence) of AS is allowed. The critical supersaturation of AS is not taken as a fixed value in terms of salt molality; rather, it is determined based on the molal ion activity product (IAP) of AS, which is defined as

$$\text{IAP}_{AS} = [a_{\text{NH}_4^+}^{(m)}]^2 \, [a_{\text{SO}_4^{2-}}^{(m)}]^1, \tag{2}$$

where $a_{\text{NH}_4^+}^{(m)}$ and $a_{\text{SO}_4^{2-}}^{(m)}$ are the molal activities of the ammonium and sulfate ions in solution (Zuend et al., 2010). In other words, the non-ideal interactions in the liquid solution have an effect on $\text{IAP}_{AS}$. This is consistent with classical nucleation theory in that the molar Gibbs energy difference (energy barrier) required for efficient nucleation of a crystalline AS phase at a certain temperature (e.g., Gao et al., 2006) is related to a critical value of $\text{IAP}_{AS}$. Generally, the process of nucleation-and-growth of a new crystalline phase from a liquid salt solution is stochastic in nature, yet the number of nucleation events per unit time and volume increases exponentially once a characteristic energy barrier is overcome as the supersaturation increases. In this study, the critical value of $\text{IAP}_{AS}$ at the point of crystallization is taken (in a deterministic manner) as

$$\text{IAP}_{AS}^{(crit.)} = c_{AS} \times \text{IAP}_{AS}^{(sat.)}. \tag{3}$$

Here, $\text{IAP}_{AS}^{(sat.)}$ is the molal ion activity product of AS at salt saturation computed by AIOMFAC for the aqueous AS system at a temperature of 298.15 K, for which the molality of AS is known from bulk measurements: $m_{AS}^{(sat.)} = 5.790$ (Apelblat, 1993). The multiplication factor $c_{AS}$ is taken as a constant coefficient relating the IAP at AS saturation to the one at crystallization in small suspended solution droplets. We determined an approximate value of $c_{AS} = 28$ by matching AIOMFAC calculations of AS

molality-dependent water activity to observed efflorescence RH obtained from electrodynamic balance (micrometer-sized aqueous AS droplets) and hygroscopicity tandem differential mobility analyzer (HTDMA; submicrometer-sized AS droplets) measurements (Zardini et al., 2008). Close to room temperature (~290 - 298 K), such experiments show that the phase transition of AS crystallization typically occurs in the range from 35% - 40% RH (Zardini et al., 2008; Ciobanu et al., 2010). With this procedure, the crystallization point (and efflorescence RH) of AS can be calculated for any mixture containing AS. In the case of the aqueous PEG-AS droplets, a LLPS is predicted to be present in the RH range where AS crystallizes during a dehydration experiment. This LLPS leads to AS partitioning to a predominantly aqueous AS phase that consequently shows crystallization at approximately the same RH as in the case of the binary water + AS system.

As discussed by Hodas et al. (2015), the AIOMFAC-based model predicts phase compositions (including water), which allow for a straightforward calculation of hygroscopic mass growth factors. However, to obtain a diameter growth factor for direct comparison with the DASH-SP-determined HGFs, knowledge of the mixture density or of the partial density or volume contributions by the mixture components are necessary for the conversion. Here we assume that the particles are spherical in shape and that the partial volumes of the mixture components in the liquid phases are additive. Densities of ammonium sulfate in the solid and liquid state were obtained from Clegg and Wexler (2011). A value of $\rho_{PEG200} = 1.121$ g cm$^{-3}$ is used for the liquid-state density of pure PEG200 at 298.15 K based on tabulated data by Ayranci and Sahin (2008). For pure PEG1000 and PEG10000, the (subcooled) liquid-state densities at 298.15 K were calculated based on tabulated data and density model coefficients by Mohsen-Nia et al. (2005). The values used are 1.1737 g cm$^{-3}$ for PEG1000 and 1.185 g cm$^{-3}$ for PEG10000.

## 2.5 $\kappa$-Köhler theory and computation of the hygroscopicity parameter $\kappa$

The hygroscopicity of single solutes, and the effective hygroscopicity of mixtures of components, is commonly parameterized with $\kappa$-Köhler theory (Petters and Kreidenweis, 2007), in which a single parameter, $\kappa$, is introduced to account for the solute effect on particle water uptake and CCN activation. Ideally, this parameter accounts for all solute effects on water activity and hence replaces the water activity factor in the Köhler equation by an expression based on $\kappa$. The Köhler equation (Eq. 4) describes the equilibrium water saturation ratio, $S$, over a curved droplet as function of droplet (wet particle) diameter $D = D_0$ HGF, air-droplet surface tension σ, temperature $T$, and water activity $a_w$ of the droplet solution (e.g. Petters and Kreidenweis, 2007).

$$S = a_w \exp\left[\frac{4\,\sigma M_w}{RT\rho_w D_0\,\text{HGF}}\right], \tag{4}$$

where $M_w$ and $\rho_w$ are the molar mass of water and the density of pure water in the liquid state at $T$, respectively, and $D_0$ is the reference (dry) particle diameter at 0 % RH. Based on the definition of $\kappa$ by Petters and Kreidenweis (2007) for non-volatile solutes and by using HGF to express the water content (mixture composition), the following expression is obtained for $\kappa$ at a certain HGF ($\kappa_{HGF}$):

$$\kappa_{HGF} = 1 - HGF^3 + \frac{HGF^3 - 1}{S} \exp\left[\frac{4\,\sigma M_w}{RT\rho_w D}\right] \tag{5}$$

This expression can be used directly to compute effective values of $\kappa_{HGF}$ from measured particle diameters at a set saturation ratio (instrument RH setpoint). By substituting Eq. (4) for $S$ in Eq. (5) we obtain the following expression that directly links $\kappa_{HGF}$ and water activity:

$$\kappa_{HGF} = 1 - HGF^3 + \frac{HGF^3 - 1}{a_w} = (HGF^3 - 1)\left(\frac{1}{a_w} - 1\right) \tag{6}$$

This equation is of use for the direct calculation of a mixture's $\kappa_{HGF}$ with a thermodynamic model, in which water activity and HGF are both computed based on mixture composition and pure component densities. Herein, the AIOMFAC-based equilibrium model was used to predict Köhler curves as well as $\kappa_{HGF}$ values as a function of RH ($S$) for a given set of assumptions about the initial dry size of the particles

and various droplet surface tensions (see section 3.3). To allow for discussion regarding differences in hygroscopicity above and below water saturation, values of $\kappa$ at CCN activation ($\kappa_{CCN}$) were also determined for the aerosol systems studied here. Values of $\kappa_{CCN}$ were defined as the $\kappa$ value at the particle maximum in particle equilibrium supersaturation, which corresponds to the maximum in the Köhler curve for that aerosol system.

As is commonly done for field and laboratory experiments, values of $\kappa_{HGF}$ (at RH = 90%) and $\kappa_{CCN}$ were also calculated based on experimental observations using Eq. (5) and the following approximate equation, respectively, (Petters and Kreidenweis, 2007):

$$\kappa_{CCN,} \approx \frac{4}{27}\left(\frac{4\sigma M_w}{RT\,\rho_w\,D_{crit}}\right)^3 [\ln(S_c)]^{-2} \tag{7},$$

with critical saturation ratio, $S_c$, set to the CCN counter saturation ratio set point (1.008 here) and dry

critical diameter, $D_{crit}$, derived from experimental observations as described above. For calculations of $\kappa$ based on experimental observations, results indicate apparent hygroscopicity, as it is assumed that droplet surface tension is that of water and, thus, this parameterization of hygroscopicity considers the combined effects of surface tension and solubility. As a result, we refer to values of $\kappa$ derived from experimental measurements as $\kappa_{HGF,app}$ and $\kappa_{CCN,app}$. In addition deviations from the assumption of dilute conditions for

aerosol systems with $\kappa_{CCN} < {\sim}0.2$ can contribute to uncertainty in values of $\kappa_{CCN}$ obtained with Eq. (7) (Petters and Kreidenweis 2007).

**2.6 Estimation of water diffusivity and mixing timescale**

The extent to which kinetic limitations may have contributed to the observed water uptake was further explored using the bulk diffusivity estimation scheme presented in Berkemeier et al., (2014). The bulk diffusivity of water ($D_{H2O}$) and characteristic timescale of bulk diffusion ($\tau_{cd}$), a metric of the time required

for particles to achieve equilibrium with water vapor, were calculated at RHs ranging from 0 to 100% and at temperatures of 253 to 298 K for 250 nm particles comprised of PEG200 and PEG10000. Knowledge of

$\tau_{cd}$ at 298 K provides insight into whether humidification equilibrium was likely to be reached during HGF measurements. Diffusivity estimations at temperatures < 298 K explore the influence of viscous aerosol components on water uptake at the range of temperatures experienced by particles in the troposphere, including those relevant to cloud base height. During particle hydration, the diffusion of water through a non-supersaturated aqueous PEG solution would be more representative, due to the partial dissolution of the PEG-containing aerosols with increasing RH. Modeled conditions are more relevant to diffusion through a PEG shell, which may form during the rapid drying of the particles studied here. Thus, modeled water diffusion in pure PEG will likely represent a lower limit for diffusivity.

$D_{H2O}$ in aqueous PEG systems is parameterized based on the Vogel-Fulcher-Tamman description of the behavior of glass-forming liquids (Vogel, 1921; Fulcher, 1925; Tammann and Hesse, 1926):

$$D_{H2O}(T, a_w) = 10^{-\left(A(a_w) + \frac{B(a_w)}{T - T_0(a_w)}\right)}, \tag{8}$$

where $T$ is temperature, $a_w$ is mole fraction-based water activity, $A$ is the high-temperature maximum of $D_{H2O}$, $B$ is a parameter describing the functional form of diffusivity as $T_g$ is approached, and $T_0$ is the temperature at which $D_{H2O}$ approaches zero (Berkemeier et al., 2014). $A(a_w)$ and $B(a_w)$ can be estimated based on parameterizations for a chemically-similar reference compound, here sucrose (Zobrist et al., 2011), relying on the assumptions that (1) $D_{H2O}$ of PEG and sucrose behave similarly approaching $T_g$ (i.e., $B_{sucrose} \approx B_{PEG}$), (2) $D_{H2O}$ is similar for PEG and sucrose at the high temperature limit (i.e., $A_{sucrose} \approx A_{PEG}$), and (3) both systems exhibit a similar ratio of $T_0$ to $T_g$ (i.e., $T_{0,sucrose}/T_{0,PEG} \approx T_{g,sucrose}/T_{g,PEG}$) (Berkemeier et al., 2014). Values of $a_w$ and $T_g$ for PEG200 and PEG10000 were taken from the literature (Ninni et al., 1999; Pielichowsk and Flejtuch, 2002; Dow et al., 2011). Calculated values of $D_{H2O}$ were then used to estimate the $e$-folding time of bulk diffusion (i.e., the time required for the concentration of water in the core of a particle exposed to a given RH to be within a factor of $1/e$ of thermodynamic equilibrium) for 250 nm particles comprised of both PEG200 and PEG10000:

$$\tau_{cd} = \frac{d_p^2}{4\pi^2 D_{H2O}}, \tag{9}$$

where $d_p$ is particle diameter and all other parameters are as defined above (Shiraiwa et al., 2011).

## 3 Results and discussion

### 3.1 Measurements of hygroscopic growth and CCN activity

Measured HGFs for the PEG and PEG-AS systems are shown in Figure 1. For the systems containing only PEG, particles displayed moderate growth: HGFs were 1.35 for PEG200 and 1.30 for both PEG1000 and PEG10000 at an RH of 90%, 23% and 26% lower than the HGF measured for the pure AS aerosols. These HGFs correspond to $\kappa_{HGF,app}$ values of 0.162 and 0.133. In agreement with results of previous studies of the hygroscopicity of polymers and HULIS, we observed little variability in hygroscopic growth across PEG systems with different molecular masses (Brooks et al., 2004; Petters et al., 2006; Ziese et al., 2008). HGFs

measured here are within ~5% of those measured for 100 nm (dry size) particles comprised of PEG with average molecular masses of 600 and 3,400 measured by Petters et al., (2006). In general, results were similar for the PEG-AS systems; however, at RHs of 80 and 90%, greater growth was observed for PEG200-AS particles (HGFs = 1.39 and 1.60) compared to PEG1000-AS (HGFs = 1.24 and 1.38) and

PEG10000-AS particles (HGFs = 1.35 and 1.42). HGFs at RH = 90% are 9%, 21%, and 19% lower than that for pure AS for PEG200-AS, PEG1000-AS, and PEG10000-AS, respectively, and correspond to $\kappa_{HGF,app}$ values of 0.344, 0.181 and 0.207. While HGFs for the PEG10000-AS aerosols exceed those for the PEG1000-AS system at RHs of 80 and 90%, this difference is within experimental uncertainty. The large degree of similarity in HGFs for the aerosol systems containing PEG1000 and PEG10000, despite the order

of magnitude difference in molecular mass, is in agreement with previous hygroscopic-growth measurements showing that the influence of polymer chain length on water uptake displays a threshold behavior, becoming relatively constant for higher degrees of polymerization (Baltensperger et al., 2005; Petters et al., 2006).

In contrast with the HGF measurements, measurements of CCN activity, as characterized by $D_{crit}$

at a constant supersaturation of 0.8%, suggest that particle hygroscopicity at conditions of high RH increases with increasing molecular mass of the PEG oligomer (since the measurements suggest a decrease in $D_{crit}$ with increasing PEG molar mass). CCN activity of the PEG-only containing systems was significantly diminished compared to the AS control, with $D_{crit}$ values of 65.4, 63.9, and 61.6 nm for PEG200, PEG1000, and PEG10000 respectively, compared to an activation diameter of ~32.7 nm for AS

(Figs. 2a, 3). Values of $D_{crit}$ for PEG200, PEG1000, and PEG10000, correspond to $\kappa_{CCN,app}$ values of 0.076, 0.082, and 0.091. For PEG200, the value of $\kappa_{CCN,app}$ derived here from CCN measurements falls well within the range previously reported for particles comprised of Tetraethylene glycol and Pentaethylene glycol (molecular masses = 194 and 238 g/mol, respectively; $\kappa_{CCN,app}$ = 0.057 – 0.195) (Petters et al., 2009b). For PEG1000, the $\kappa_{CCN,app}$ observed here is higher than the upper limit of the range reported by

Petters et al., (2009b) (0.033 – 0.064), but lower than the upper limit reported in that study for PEG400 (0.05 – 0.106). Potential contributors to differences in estimates of $\kappa_{CCN,app}$ for PEG1000 include differences in experimental conditions for the CCN measurements (e.g., particle size and/or supersaturation), differences in the proportions of PEG oligomers with varying chain lengths resulting in an average molecular mass of 1000 g/mol, as well as measurement uncertainty.

The increase in CCN activity with increasing PEG molecular mass is more clearly evident for the mixed PEG-AS systems. Critical activation diameters were 50.0, 41.4, and 28.2 nm for the PEG200-AS, PEG1000-AS, and PEG10000-AS systems, respectively (Figs. 2b, 3), corresponding to $\kappa_{CCN,app}$ values of 0.171, 0.302, and 0.953. A $\kappa_{CCN,app}$ value of 0.617 was calculated for the AS control, in agreement with previous estimates of AS hygroscopicity based on CCN measurements and κ-Köhler theory (Petters and

Kreidenweis, 2007). Differences in $D_{crit}$ for PEG200, PEG1000, and PEG10000 are within measurement uncertainty (Fig. 3), assuming an average uncertainty in DMA-transmitted particle diameter of 3.5% (Kinney et al., 1999). However, Kolmogorov-Smirnoff tests (Matlab R2014b) comparing the fitted

sigmoidal curves across the aerosol systems indicate that the full activation fraction distributions are statistically significantly different at a 95% confidence limit for PEG200 and PEG10000 (p < 0.001) and for PEG1000 and PEG10000 (p = 0.002). Differences in activation fraction distributions for PEG200 and PEG1000 are not statistically significant (p = 0.075). Significant differences in CCN activity with increasing PEG molar mass are observed for the PEG-AS aerosol systems, both in terms of the Kolmogorov-Smirnoff test on full activation fraction distributions (p = <0.001 – 0.007) and the consideration of particle-diameter measurement uncertainty. The CCN activity of the mixed PEG10000-AS particles appears to be greater than that for particles comprised only of AS, as is evident from the 15% smaller $D_{crit}$ for PEG10000-AS particles as compared to the AS control (Fig. 3). CCN activation fraction distributions for the AS and PEG10000-AS systems are significantly different (p = 0.014) and differences in $D_{crit}$ are outside of experimental uncertainty.

Observed increases in CCN activity with molecular mass and the enhancement in the CCN activity of PEG10000-AS compared to pure AS can likely be attributed to the fact that larger PEG oligomers are surface active and have been shown to lower the surface tension of the air-water interface when present in aqueous solution. There is evidence for decreases in surface tension with increasing PEG molecular mass (Rey and May, 2010; Winterhalter et al., 1995). Our results are also in agreement with previous studies that suggest that the presence of high molecular mass species (e.g., HULIS, polcycarboxylic acids) in atmospheric aerosol can contribute to decreases in surface tension (Facchini et al., 2000; Ziese et al., 2008). Sareen et al., (2013) observed that the reactive uptake of methylglyoxal and acetaldehyde resulted in enhancements in the CCN activity of AS of similar magnitude to our results for the PEG10000-AS particles (6% and 10% for methylglyoxal and acetaldehyde, respectively). In that work, the dependence of the enhancement of CCN activity on the timescales of AS exposure to organic vapors suggested that the formation of oligomers near the particle surface was a potential contributor to this effect (Sareen et al., 2013).

Our experimental results suggest a shift in the influence of molecular mass of PEG on hygroscopicity when transitioning from subsaturated RH conditions to supersaturated conditions, with a decrease in hygroscopic growth below water saturation with increasing molecular mass, but an increase in CCN activity with increasing molecular mass of PEG. Further, a marked increase in apparent hygroscopicity, as parameterized based on $\kappa_{HGF,app}$ and $\kappa_{CCN,app}$, is observed under supersaturated conditions as compared to subsaturated conditions for PEG1000-AS and PEG1000-AS. As noted above, for the purpose of comparing apparent hygroscopicity of these aerosol systems above and below water saturation, values of $\kappa$ were calculated assuming droplet surface tension is that of pure water; this assumption may not be valid, particularly for the larger molecular mass PEG oligomers. The increased influence of surface tension on water uptake under supersaturated conditions (Wex et al., 2008) is likely to be an important contributor to differences in values of $\kappa_{CCN,app}$ and $\kappa_{HGF,app}$ above and below water saturation for the PEG1000-AS and PEG10000-AS aerosol systems. In addition, differences in dry particle diameters between HGF and CCN measurements may contribute to differences in apparent hygroscopicity above and

below water saturation for all aerosol systems studied here. Potential contributors to differences in $\kappa_{HGF,app}$ and $\kappa_{CCN,app}$ are further explored below and in section 3.3.

There are several potential factors that may have contributed to our experimental results. First, it is possible that below water saturation, all systems experienced kinetic limitations to water uptake, preventing the particles from achieving equilibrium growth in the 4 s residence time of the DASH-SP humidifiers, potentially contributing to similarities in hygroscopic growth across the systems. As water content increases, however, particle viscosity is expected to decrease due to the plasticizing effect of water. While potentially explaining differences in hygroscopicity below and above water saturation for the systems that do exhibit enhanced apparent hygroscopicity under supersaturated conditions, this does not explain why CCN activity would increase with PEG molecular mass. It has been suggested that hygroscopic growth of some semisolid particles may proceed via adsorption below water saturation, but via absorption at RHs relevant to CCN activation (Pajunoja et al., 2015). In addition, the magnitude of the influence of non-ideal interactions between aerosol components, as well as other factors that influence water uptake, may differ under the concentrated conditions relevant to subsaturated hygroscopic growth as compared to water uptake in supersaturated environments. We further investigated the potential influence of these factors on experimental results by comparing measurements to predictions from the AIOMFAC model and estimations of water diffusivity and characteristic time of bulk diffusion.

**3.2 Comparison of HGFs from the DASH-SP and the AIOMFAC model**

Comparison between HGFs measured with the DASH-SP and those calculated with the AIOMFAC-based model at thermodynamic equilibrium provides a means to explore the potential influence of kinetic limitations to water uptake on observed hygroscopic growth. For the sake of brevity in the following discussion, the AIOMFAC-based equilibrium model predictions (described in Section 2.4) are simply referred to as "AIOMFAC predictions;" however, we note that, more precisely, the AIOMFAC model is just the core part of the equilibrium model that computes activity coefficients (and activities). Experimental results are generally in good agreement with AIOMFAC-calculated HGFs (Figs. 4, 5). Because the model calculates thermodynamic equilibrium, this agreement suggests that kinetic limitations to water uptake did not strongly influence HGFs for the PEG-containing systems explored here. As noted above, if particles did not achieve equilibrium with water vapor in the ~5 s residence time of the DASH-SP and diffusion dryers and/or in the 4 s residence time of the DASH-SP humidifier, it would be expected that experimental observations would deviate substantially from growth curves predicted by AIOMFAC. AIOMFAC predictions of water uptake for the PEG systems are also in excellent agreement with experimental bulk water activity data for these systems (Appendix A, Fig. A2), suggesting that agreement between DASH-SP measurements and AIOMFAC-based predictions also indicate that measured HGFs are consistent with bulk water activity measurements. Model-measurement disagreement at an RH of 80% for the PEG200-AS and

PEG10000-AS systems may result from uncertainty in PEG-specific AIOMFAC parameterizations and/or uncertainty in measured HGFs just above the deliquescence of AS.

The generally good agreement between AIOMFAC and HGFs measured with the DASH-SP also suggests that equilibrium absorption sufficiently describes the water-uptake behavior of the PEG aerosol systems at RHs below water saturation. Thus, it is unlikely that differences in the mechanisms of growth (i.e., adsorption versus absorption) explain the discrepancies in the influence of molecular mass on subsaturated hygroscopic growth and CCN activity observed here. This adds support to the conclusions of Pajunoja et al. (2015) that low solubility, rather than viscosity and related kinetic limitations, drives adsorption-dominated growth at low RH for some semisolid organic aerosol constituents. While the solubility of PEG in water decreases with increasing molecular mass, PEG for all molecular masses studied here is highly water soluble (Dow et al., 2011). While unlikely to be a major factor here, it is important to note that uncertainty in HGFs obtained with the DASH-SP, and potentially other measurement methods that derive hygroscopic growth from light scattering by particles, may be larger for systems that do experience adsorptive growth as compared to those for which water uptake is driven by absorption at all RHs. This is because calculations of HGF from DASH-SP OPC pulse heights rely on the assumption that particle refractive index can be represented as a volume-weighted average of the refractive indices of water and that of the dry particle components. This assumption may not be valid under conditions in which adsorptive growth dominates and a layer of water on the particle surface influences particle optical properties.

AIOMFAC predictions of the phase states of the aerosol systems and the prevalence of LLPS may also provide insight into other aspects of observed hygroscopic growth (Fig. 5). In particular for aerosol systems that undergo LLPS, differences in the RH at which two separated liquid phases merge to a single phase might explain the greater growth observed for the PEG200-AS system as compared to the PEG1000-AS and PEG10000-AS systems at RH of 90%. For the PEG200-AS system, AIOMFAC predicts the merging of the AS-dominated and PEG-dominated phases at an RH ~ 86%, while for PEG1000-AS, LLPS is predicted to persist up to RH ≈ 94% (Fig. 5). With the merging of the phases for the PEG200-AS system, it is possible that more water is taken up when PEG and AS are present in a single phase than would be expected assuming that the two components take up water independently. At an RH of 90%, the model-predicted HGF for the single-phase liquid mixture comprised of PEG200 and AS is ~4% greater than the HGF calculated with a ZSR-type mixing rule in which water uptake by the individual aerosol components is treated separately (with total water uptake calculated in an additive manner), suggesting that this may have a small impact on observed HGFs at high RH. As noted above, complete separation between PEG10000 and AS was assumed at all RH values for the PEG10000-AS system. The validity of this assumption is supported by modeling results for PEG200-AS and PEG1000-AS, which suggest that the RH at which separated phases merge to a single phase increases with increasing PEG molecular mass.

**3.3 Predicted hygroscopicity parameter, Köhler curves and CCN activation**

To further explore contributors to differences in water-uptake behavior across aerosol systems, as well as differences in apparent hygroscopicity under sub- and supersaturated conditions within individual aerosol systems, we performed AIOMFAC calculations with a high resolution for the high-RH range above 90% RH toward 100 % RH with respect to bulk solution systems. Using the Köhler equation (Eq. 3), the computed water activity and HGF data can then be used to obtain Köhler curves, as well as $\kappa_{HGF}$ and $\kappa_{CCN}$ as a function of saturation ratio for particles of specified dry sizes and assumed surface tensions (Figs. 6 and 7). As noted in section 2.6, values of $D_{crit}$ were calculated based on Köhler theory assuming a range of droplet surface tensions from the value of 72 mN m$^{-1}$ for pure water (at ~298 K) to much lower values typical for PEGs and mixtures of PEGs with ammonium sulfate (Wu et al., 1996; Song et al., 2013). We then compared calculated values of $D_{crit}$ to values of $D_{crit}$ derived from CCN measurements to evaluate the droplet surface tension that provided the best agreement with measurements (Fig. 3). Based on this comparison, we performed calculations of Köhler curves and $\kappa_{HGF}$ for particles of dry size $D_0 = 50$ nm and three different values for the surface tension, $\sigma$,: 72, 50, and 40 mN m$^{-1}$. In general, the actual surface tension of a particle may depend in a nonlinear fashion on its bulk composition and potential bulk-to-surface partitioning, such as surface enhancement of less polar components (Sorjamaa et al., 2004; Ruehl et al., 2016). Here we use three chosen, fixed values for $\sigma$ in a reasonable numeric range based on the observed $D_{crit}$ behavior (Fig. 3). This allows for a discussion of the influence of surface tension on predicted CCN activity alongside with calculated $\kappa_{HGF}$ at high RH.

Figures 6 and 7 show the predicted Köhler curves and the hygroscopicity parameters $\kappa_{HGF}$ for the different single-solute systems and the PEG-AS mixtures. The $\kappa_{HGF}$ are shown against the RH in equilibrium with the 50 nm particles (i.e., for RH = $S$ based on Eq. 3) and also for a bulk mixture, for which the Kelvin (curvature) effect is absent and for which water supersaturation is therefore not reached. Calculated values of $\kappa_{CCN}$, the $\kappa_{HGF}$ value at the particle maximum in equilibrium supersaturation corresponding to the maximum in the Köhler curve for that particle, are also shown. AIOMFAC-predicted values of $\kappa_{CCN}$ for the PEG200 and PEG1000 aerosol systems (0.108 and 0.037, respectively for an assumed surface tension of 72 mN/m) are in very good agreement with the range of values previously reported for Tetra- and Pentaethylene glycol (comparable to PEG200) and for PEG1000 (Petters et al., 2009b).

With the exception of the pure AS system (Fig. 6h), all PEG and mixed PEG-AS mixtures exhibit a decrease in calculated values of $\kappa_{HGF}$ with increasing RH, regardless of the choice of surface tension value. This indicates that the systems actually become less hygroscopic with increasing RH at high RH in terms of the single parameter concept of $\kappa$-Köhler theory. Also, the finding that all $\kappa_{HGF}$ values vary with RH illustrates a limitation of using $\kappa_{HGF}$ obtained from HGF measurements at a certain RH (typically at around 80 % to 90 %) to infer the hygroscopic growth behavior at high RH and particularly for supersaturated conditions. Recent laboratory studies have indicated that SOA formed from the oxidation products of α-pinene display the same behavior of decreasing $\kappa_{HGF}$ with increasing RH (Pajunoja et al., 2015; Renbaum-Wolff et al., 2016). The results here indicate the power of models like AIOMFAC to

bridge that gap in RH range and – given reasonably good agreement observed between modeled and measured HGFs under subsaturated conditions – to be applicable for the prediction of the CCN activity for given (known or estimated) supersaturations (or dry particle size) and surface tension values.

A comparison of the observed $D_{crit}$ values for PEG1000-AS (41.4 nm at 0.8% supersaturation) and
PEG200-AS (50.0 nm) indicates that either the hygroscopicity in the mixture with PEG1000 must increase at supersaturated conditions, as discussed above, or that the surface tension is significantly lower in comparison to the PEG200-AS mixture to explain the lower critical activation diameter measured for PEG1000-AS with the same dry PEG:AS mass ratio in both mixtures. Both effects (higher hygroscopicity or lower surface tension) could explain an increase in the CCN activity at a given supersaturation. In the
calculations shown in Figures 6 and 7, the supersaturation is not fixed; rather, the dry particle size is, which then leads to different supersaturations for cloud droplet activation as indicated by the maxima in the Köhler curves (which depend considerably on the chosen value for surface tension). For any choice of the same surface tension for each system, the critical supersaturation for CCN activation of the PEG1000-AS system is predicted to be slightly higher than that for PEG200-AS. In a similar sense, the predicted $\kappa_{CCN}$
values are higher in the case of the PEG200-AS particles (~0.231 to 0.234 for PEG200-AS and ~0.172 to 0.175 for PEG1000-AS). This means that in terms of the parameter $\kappa$, the PEG200-AS system is considered more hygroscopic both at 90 % RH and at supersaturated conditions, in agreement with the measured and modeled HGF at subsaturated conditions. However, the $D_{crit}$ values derived from CCN measurements for PEG1000-AS are slightly smaller, which is therefore a trend against the predicted hygroscopicity. This
indicates that the hygroscopicity (i.e., the "solute effect" in Köhler theory) does not explain the observed CCN activity and instead suggests that the surface tension of the PEG1000-AS droplets must be lower than that of the PEG200-AS droplets. Indeed, the comparison in Figure 3 suggests that the surface tension of the PEG200-AS droplets is close to the value for pure water droplets, while σ ≈ 60 mN m$^{-1}$ for PEG1000-AS. Such a change in surface tension is well within the range of droplet surface tensions observed for high
molecular weight species such as HULIS (Kiss et al., 2005; Salma et al., 2006; Taraniuk et al., 2007).

The smaller hygroscopicity of PEG1000-AS compared to PEG200-AS means that the aqueous solution is more concentrated in the PEG1000-AS system at high RH close to CCN activation (lower water content), which may then lead to a lower surface tension than that of pure water since the surface composition is likely as concentrated or more concentrated in PEG1000 than the droplet bulk, even when
no enhanced bulk-to-surface partitioning by PEG1000 is assumed. The combined effect of moderate hygroscopicity and lowering of the droplet surface tension is likely also present, and potentially enhanced, in the case of the PEG10000-AS system. For the PEG10000 and PEG10000-AS aerosol systems, agreement between $D_{crit}$ derived from CCN measurements and those calculated using Köhler theory was achieved for assumed values of σ of between 40 and 50 mN/m and of 40 mN/m, respectively (Fig. 3), in
agreement with previously published values of σ for PEG10000 (~ 43 - 45 mN m$^{-1}$) (Wu, 1974; Wu et al., 2011).

Panel (d) of Figure 7 shows that one characteristic of a phase transition from an LLPS region to a single liquid phase with increasing particle water content (increasing RH) is a kink in the hygroscopic growth curve and in associated $\kappa_{HGF}$ (in mathematical terms: a "removable" discontinuity in curve smoothness). The vertical yellow line in that figure shows the particle diameter or RH of phase transition for the bulk system (red curve; Fig. 7c,d). The LLPS onset/offset for the 50 nm particles is shifted toward higher RH (as the same HGF shifts upwards in saturation ratio with decreasing particle size) and is visible by the corresponding kinks in those curves. We note that this upward shift in RH for LLPS onset may be counteracted in cases where the composition for LLPS onset itself becomes a size-dependent property (due to a penalty in Gibbs energy for forming the liquid-liquid interface in small particles).

High-resolution calculations of phase behavior at high RH also provide further insight into experimental results and indicate an influence of the persistence of LLPS (in terms of RH space) on water-uptake behavior. As noted above, for the PEG200, PEG1000, and PEG10000 aerosol systems, for which no phase separation was predicted, values of $\kappa_{CCN,app}$ are lower than values of $\kappa_{HGF,app}$ calculated from measurements conducted below water saturation. Similarly, for the PEG200-AS system, for which separated phases are predicted to merge to a single liquid phase at RH < 90% (Fig. 5), a small decrease in $\kappa_{CCN,app}$ compared to $\kappa_{HGF,app}$ was observed. On the other hand, we observed substantial enhancements in apparent hygroscopicity under supersaturated conditions as compared to subsaturated conditions for the PEG1000-AS and PEG10000-AS systems. As noted above, model predictions suggest that the RH at which separated phases merge to a single phase increases with PEG molecular mass, and it is expected that LLPS persists up to RH values near 96% (and potentially beyond that) for the PEG10000-AS system. A recent study suggests that discontinuities in water-uptake behavior below and above water saturation for SOA formed from the oxidation products of α-pinene could be attributed to the presence of LLPS at RH values approaching those relevant to CCN activation (RH > 95%) (Renbaum-Wolff et al., 2016). The prevalence of LLPS at subsaturated conditions for these systems is an indication for limited miscibility among the mixture components. This likely affects the bulk-to-surface partitioning at high RH, with the PEG molecules having a higher thermodynamically-driven affinity for partitioning to the air droplet interface due to their lower pure-component surface tensions (Song et al, 2013). Thus, differences in bulk-to-surface partitioning are expected to contribute to differences in droplet surface tensions and observed CCN behavior.

The influence of surface tension on observed CCN activity can be further explored based on the concept of the presence of an organic film of a certain thickness, $\delta_{org}$, at the air-droplet interface (Ruehl et al., 2016). Ruehl et al. (2016) introduced a compressed film model to relate surface tension depression by surface-active organic aerosol components at sub/supersaturated conditions to organic surface coverage or apparent organic film thickness (Ruehl et al., 2016). The parameter $\delta_{org}$ approximates the thickness of the organic film at CCN activation based on the assumption that all or a portion of organic material is adsorbed to the surface instead of being dissolved in the droplet bulk, with a thinner film (i.e., smaller $\delta_{org}$) indicating enhanced CCN activity. For a simplified case in which it is assumed that all PEG material is forming a

surface film, values of $D_{crit}$ for PEG200-AS, PEG1000-AS, and PEG10000-AS derived from CCN measurements correspond to $\delta_{org}$ values of 0.35, 0.17, and 0.05 nm, respectively. For mixtures of AS with surface-active dicarboxylic acids, for which enhanced CCN activity was observed, Ruehl et al. (2016) found that $\delta_{org}$ ranged from 0.07 to 0.21 nm. The similarities in these ranges of assumed film thickness for PEG1000-AS and PEG10000-AS further support the conclusion that PEG contributes to the CCN activity of these mixed inorganic-organic particles through its surface activity and influences observed discontinuities in apparent hygroscopicity below and above water saturation for these systems.

### 3.4 Water diffusivity in PEG and characteristic equilibration timescales

Calculations of $D_{H2O}$ and $\tau_{cd}$ provide further insight into the extent to which viscosity-induced limitations to the mass transport of water in PEG may have contributed to differences in water-uptake behavior below and above water saturation. Specifically, these calculations allow us to explore whether the diffusivity of water in PEG is sufficiently slow at low and moderate RH levels that timescales for the particles to achieve equilibrium with water vapor exceed the 4 s residence time of the DASH-SP humidifier. Limitations to the mass transport of water in the particle bulk are expected to decrease with increasing RH and particle water content because water serves as a plasticizer for viscous aerosol components. As noted above, if estimations of water diffusivity and mixing timescales do suggest substantial kinetic limitations to hygroscopic growth under subsaturated conditions, this could explain, at least in part, differences in hygroscopicity below and above water saturation, but not observed increases in CCN activity with increasing PEG molecular mass.

In line with the agreement between DASH-SP and AIOMFAC HGFs, the minimal impact of kinetic limitations to water uptake on measured HGFs is supported by calculations of $D_{H2O}$ and $\tau_{cd}$ (Fig. 8). For the conditions under which experiments were conducted ($T = 298$ K), the diffusivity of water in both PEG200 and PEG10000 is fast enough that values of $\tau_{cd}$ are predicted to be ~$10^{-3}$ s at low RH. $\tau_{cd}$ decreases to ~$10^{-6}$ s as RH approaches 100%. Thus, the 4 s humidification timescale in the DASH-SP is sufficient to achieve equilibrium for the PEG systems studied here. It is not expected that slow diffusion of water in PEG at subsaturated RHs contributes to the observed discrepancies in water uptake behavior above and below water saturation. This is in agreement with previous results suggesting that despite mechanical behavior indicative of solid or semisolid particles, the diffusivity of small molecules (e.g., water) in SOA from a variety of precursors is high at room temperature (Shiraiwa et al., 2013; Price et al., 2015; Lienhard et al., 2015).

The diffusivity estimation results also provide insight into the conditions under which kinetic limitations driven by the inhibition of water transport in viscous aerosol components may be important. As is shown in Figure 8, at colder temperatures ($T = 253$ K) relevant to higher altitudes in the free troposphere, the diffusion of water in both PEG200 and PEG10000 is slowed and equilibration timescales approach 100

s, depending on ambient RH. Thus, while equilibrium partitioning sufficiently describes the water uptake behavior of PEG-containing aerosol systems under the experimental conditions considered here, this might not be the case for viscous aerosols under all atmospherically relevant conditions. It is expected that at colder ambient temperatures, increases in particle viscosity will result in larger discrepancies between water-uptake behavior below and above water saturation than observed in the present experiments. The study of water diffusivity, hygroscopic growth, and CCN activity of aerosols under a range of atmospherically relevant temperatures is an active area of research (e.g., Berkemeier et al., 2014; Price et al., 2015; Steimer et al., 2015; Lienhard et al., 2015).

**3.5 Phenomena contributing to discontinuities in water-uptake behavior below and above water saturation**

The supplementation of experimental results with theoretical predictions from AIOMFAC and estimations of water diffusivity suggests that observed discontinuities in the influence of PEG molecular mass on aerosol hygroscopicity under subsaturated and supersaturated RH conditions (i.e., similar growth across PEG systems at RH < 100%, but increasing CCN activity with increasing molecular mass) cannot be explained by RH-dependent particle viscosity (and associated kinetic limitations to water uptake) nor differences in the mechanisms of hygroscopic growth (i.e., adsorption versus absorption). However, the ability of AIOMFAC to successfully describe the subsaturated hygroscopic growth of the PEG aerosol systems does provide insight into the factors likely to be contributing to observed differences in water-uptake behavior below and above water saturation, as AIOMFAC explicitly accounts for non-ideal interactions between aerosol components. AIOMFAC-predicted mole fraction based activity coefficients of water in PEG200, PEG1000, and PEG10000 are shown in Figure 9. It is evident that activity coefficients are substantially lower for the higher molecular mass PEGs up to an RH ~ 95%, indicating a greater degree of non-ideality for those solutions. This suggests that the degree to which Raoult's law will under predict water uptake is greater for high molecular mass compounds, indicating the importance of accounting for molecular size in water activity models. As expected, activity coefficients of water converge towards 1.0 (ideality) as water saturation is approached. At the RH values at which HGFs were measured, non-ideal interactions between PEG and water have a substantial influence on hygroscopic growth, while under conditions relevant to CCN activity, the influence of these non-ideal interactions is expected to be negligible (at least in the droplet bulk). Thus, we conclude that observed differences in hygroscopic growth and CCN activity can be attributed, at least in part, to the greater influence of non-ideal interactions under the more concentrated conditions (i.e., lower water contents) relevant to subsaturated hygroscopic growth as compared to supersaturated conditions.

Our findings are in agreement with a sensitivity analysis conducted by Wex et al. (2008), in which the influence of various parameters in the Kohler model were evaluated across a range of atmospherically relevant RH values. At the more concentrated conditions relevant to hygroscopic growth below water

saturation, variables influencing water activity were found to dominate in driving variability in water uptake. On the other hand, the influence of surface tension was found to be negligible at RH < 95%, but to be an important determinant of growth at RH ≥ 95% and particularly for CCN activity. RH-dependent variability in the degree to which surface tension influences water uptake is another likely contributor to our experimental results. Because the effectiveness with which the PEG oligomers depress the surface tension of the air-particle interface is expected to increase with the molecular mass of the PEG polymer (Rey and May, 2010; Winterhalter et al., 1995), likely as a result of enhanced bulk-to-surface partitioning with increasing polymer size, we observed increases in CCN activity with increasing PEG polymer chain length. However, the effects of surface tension are negligible at the RH values at which HGFs were measured with the DASH-SP, contributing to the relatively similar hygroscopic behavior across the PEG aerosol systems under subsaturated conditions. Previous work has suggested that a combination of variability in the influence of surface tension and variability in activity coefficients with degree of solute dilution contributes to differences in apparent hygroscopicity based on HGF measurements and CCN activity for HULIS particles (Wex et al., 2009; Petters et al., 2009a). HULIS have been shown to have surface tensions as much as 30% lower than that of pure water at the same temperature (Kiss et al., 2005; Salma et al., 2006; Taraniuk et al., 2007).

It is important to note that in addition to RH, the influence of surface tension also varies with particle diameter. While our HGF measurements were all conducted for 250 nm particles, CCN activity was characterized based on $D_{crit}$ at a constant supersaturation. CCN activation fractions were measured for particles with diameters ranging from 20 to 210 nm. At the smaller particle sizes, the influence of surface tension is greater, also likely contributing substantially to observed enhancements in CCN activity as compared to hygroscopic growth below water saturation. This also likely explains, at least in part, why our results contrast with those of Petters et al. (2006), who found evidence for decreases in CCN activity with increasing molecular mass of polymeric species. In that study, CCN activity was characterized as the critical supersaturation for particles at a set dry size of ~100 nm. Thus, the influence of differences in droplet surface tension with increasing PEG molecular mass may not have been as evident in their measurements, as this is likely to be more important in the range of particle sizes that we studied. In addition, in that work, HGF measurements of PEG and another polymeric compound (polyacrylic acid [PAA]) were compared to CCN measurements for only PAA.

**4 Atmospheric implications**

Our results provide insight into the factors likely to be contributing to observed differences in ambient water-uptake behavior below and above water saturation. Specifically, they suggest that variability in the sensitivity of hygroscopic growth to non-ideal thermodynamic interactions and surface tension depression with RH have contributed, at least in part, to these observations and support previous work suggesting that the prevalence of LLPS at high RH contributes to differences in apparent hygroscopicity above and below

water saturation (Renbaum-Wolff et al., 2016). Notably, in many of the circumstances in which differences in sub- and supersaturated hygroscopic behavior have been observed, oligomers and other high molecular mass compounds may have been substantial contributors to total atmospheric aerosol. For example, Hersey et al. (2013) observed reductions in subsaturated hygroscopic growth but increases in CCN activity with increases in the degree of SOA aging in the eastern Los Angeles basin, where oligomeric compounds have been observed to comprise as much as 40% of submicron aerosol mass (Denkenberger et al., 2007). Differences in water-uptake behavior under sub- and supersaturated RH conditions have been observed for aerosol derived from biomass burning (Asa-Awuku et al., 2008; Dusek et al., 2011; Hersey et al., 2013). HULIS have been identified as a major component of biomass burning aerosol, and laboratory studies involving levoglucosan, a tracer of biomass burning aerosol, suggest that oligomerization reactions are likely to occur in biomass burning plumes (Holmes and Petrucci, 2006, 2007). Finally, these discrepancies in aerosol water uptake below and above water saturation have been observed in marine atmospheres (Good et al., 2010; Ovadnevaite et al., 2011). It has been hypothesized that the presence of biopolymers and biosurfactants in aerosol derived from sea-spray in biologically active waters contributes to this phenomena (O'Dowd et al., 2004; Ekström et al., 2010; Ovadnevaite et al., 2011).

Because the present work focuses only on compounds with a narrow range of chemical properties and experiments were conducted only at room temperature, other potential contributors to apparent differences in hygroscopicity above and below water saturation observed in ambient atmospheres (e.g., solubility limitations, slow diffusion of water in more viscous particles) cannot be ruled out. For example, estimates of the characteristic timescale for particles to achieve equilibrium with water vapor under different atmospherically relevant temperatures (Fig. 8) suggest that kinetic limitations to water uptake/evaporation driven by slow diffusion in viscous aerosol components may influence discontinuities in hygroscopicity above and below water saturation to different degrees depending on ambient temperature. The presence of compounds with varying solubilities that dissolve at different RHs has also been put forth as a potential explanation for enhanced CCN activity as compared to subsaturated hygroscopic growth (Petters et al., 2009a). While it is possible that due to the range of molecular masses present in each PEG reagent, lower molecular mass components dissolve into solution at lower RH values while higher molecular mass components do not dissolve until RH approached 100%, given the overall high solubility of PEG, it is unlikely that this had any substantial impact on our results. However, this phenomenon may be more important for ambient aerosol, in which compounds with a wider range of water solubilities are present.

The results of the present work also have implications for hygroscopicity measurements and the representation of aerosol hygroscopicity in large-scale atmospheric models. First, they suggest that the use of a single hygroscopicity parameter (e.g., $\kappa$) derived from HGF measurements may lead to a substantial underestimation of CCN activity in environments in which oligomers, other high molecular mass compounds, and surface-active components are present in atmospheric aerosols. In addition, due to the greater influence of surface tension depression on water uptake for smaller particles, the way in which CCN

activity is quantified may result in substantial differences in hygroscopicity characterization. For example, calculation of $D_{crit}$ from CCN activation fraction measurements conducted at a single supersaturation may result in a greater apparent hygroscopicity than if determined based on calculations of critical supersaturation, in which activation fractions of particles of a constant, typically larger diameter are measured for a range of supersaturations.

**5 Conclusions**

We observe a shift in the influence of molecular mass on the water-uptake behavior of surrogates for oligomers in atmospheric aerosol when transitioning from subsaturated to supersaturated RH conditions. For some aerosol systems, we also observe substantial enhancements in apparent hygroscopicity based on CCN measurements as compared to HGF measurements conducted below water saturation. A comparison of experimental and modeling investigations of water-uptake behavior of PEGs with a range of molecular masses and viscosities suggests that such discontinuities in apparent hygroscopicity above and below water saturation can be attributed, at least in part, to differences in the sensitivity of water uptake behavior to surface tension effects caused by enhanced bulk-to-surface partitioning of the larger PEG polymers. Under the experimental conditions investigated here, there was no evidence that kinetic limitations to water uptake due to the presence of viscous aerosol components inhibited water uptake at lower RH, nor that hygroscopic growth was driven by adsorption at low RH and absorption at high RH. This finding supports the hypothesis that limitations in solubility, rather than particle viscosity, drive the dominance of adsorptive growth at low RH observed for some semisolid particles. Enhancements in CCN activity compared to subsaturated water-uptake were evident for mixed PEG-AS aerosol systems for which LLPS is predicted to persist to up to high RH (but not above 100% RH). The prevalence of LLPS under subsaturated conditions indicates a miscibility gap due to limited solubility of PEG in concentrated aqueous AS solutions (Song et al., 2013) and is likely to influence bulk-to-surface partitioning of PEG at high RH, impacting surface droplet tension and CCN activity. The accurate description of the hygroscopic properties of particles comprised of AS and the oxidized oligomers for which PEG serves as a surrogate requires the consideration of non-ideal thermodynamic interactions between these aerosol components, including the potential presence of LLPS, as is achieved by thermodynamic models like the one based on AIOMFAC.

**Appendix A**

AIOMFAC predictions of water uptake were compared to previously published bulk water activity data for the three PEG oligomers studied here (Fig A2). Excellent agreement between AIOMFAC-predicted and measured (Ninni et al., 1999) bulk water activity data indicates that the AIOMFAC model represents the bulk diameter growth factor curves well (assuming no exotic molar volume (density) excess effects occur). Particle diameter growth factors discussed in Section 2.4 were determined from the

calculated particle mass at a certain RH with the use of pure-component densities and assumption of linear additivity of the calculated component volumes. The good agreement between AIOMFAC-predicted and measured water uptake for bulk conditions indicates that comparisons between DASH-SP measurements and AIOMFAC predictions of particle diameter growth factors, discussed in Section 3.2 are essentially equivalent to a comparison between DASH-SP measurements and measured water-uptake of bulk solutions. Figure A2 also indicates that assuming activity coefficients of unity with component mole fractions representing mixture composition would result in substantial error, as the non-ideality of aqueous PEG mixtures is large due to the pronounced difference in molecular size/mass with increasing PEG polymer chain-length compared to the size of water molecules.

*Acknowledgements.* This work was supported by the Office for Naval Research under award no. N00014-14-1-0097. Natasha Hodas was supported by a National Science Foundation Atmospheric and Geospace Sciences Postdoctoral Research Fellowship, award no. 14433246. Andreas Zuend acknowledges support by the Natural Sciences and Engineering Research Council of Canada (NSERC, grant RGPIN/04315-2014). The authors gratefully acknowledge helpful discussions with Armin Sorooshian and Taylor Shingler regarding the DASH-SP.

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

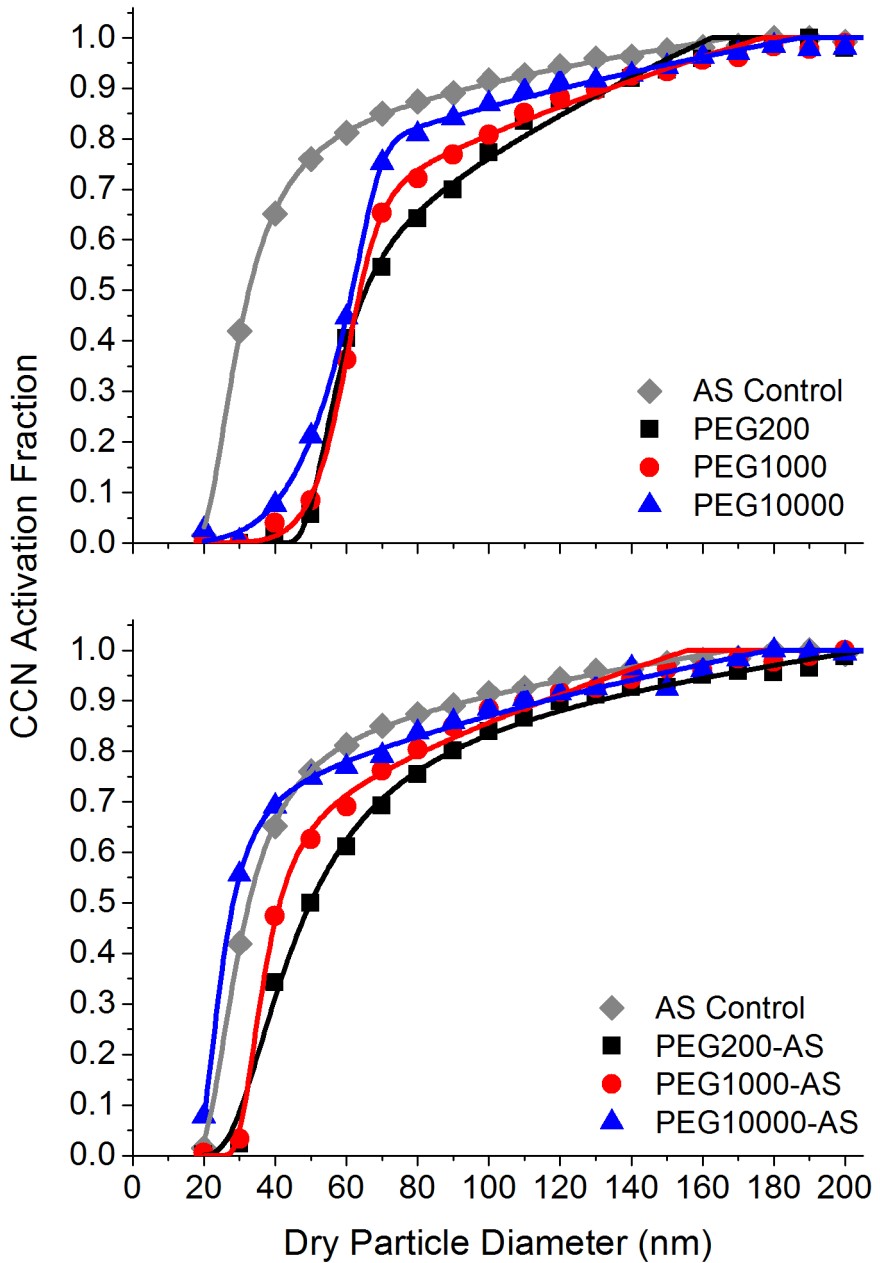

**Figure 2.** Measured cloud condensation nuclei (CCN) activation fractions as a function of dry particle size for (a) the PEG-containing aerosol systems and (b) the PEG-AS aerosol systems with PEG:AS mass ratios of 2:1. CCN activity was characterized based on the critical activation dry particle diameter at a water-supersaturation of 0.8%. Solid lines represent the sigmoidal curves fit to the activation fraction measurements for each aerosol system. Critical diameters are characterized based on the particle size corresponding to a CCN activation fraction of 50%. Pearson-r values for the sigmoidal fits are > 0.99 for all aerosol systems..

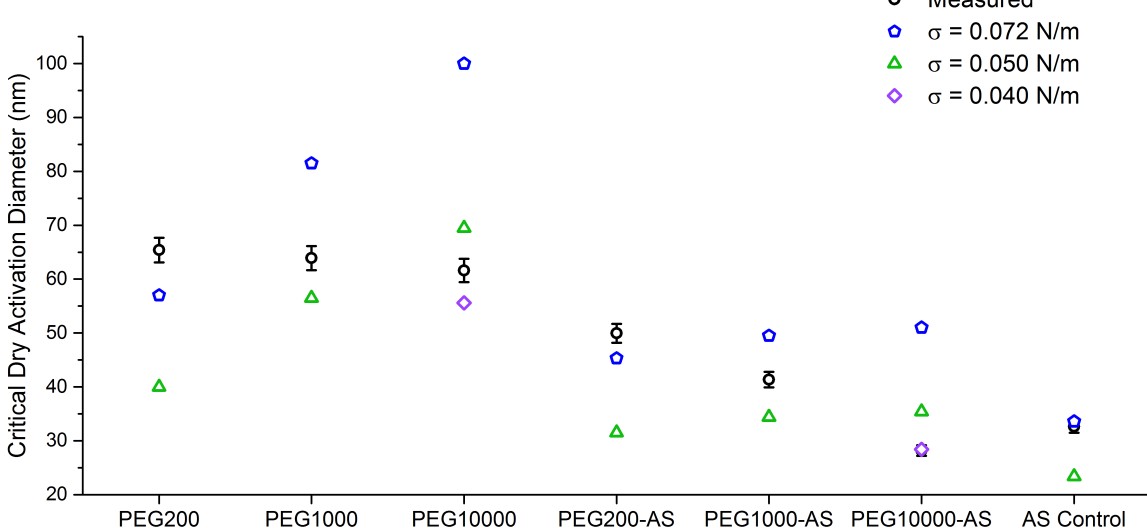

**Figure 3.** Critical dry particle diameters ($D_{crit}$) at a water supersaturation of 0.8% at 298K for the PEG, PEG-AS and, AS Control aerosol systems determined from CCN measurements (black circles) and calculated using AIOMFAC with classical Köhler theory assuming a range of droplet surface tensions ($\sigma$) Values of $D_{crit}$ were determined from CCN measurements by fitting the activation-fraction measurements with sigmoidal curves, as shown in Figure 2, and determining the dry particle diameter associated with an activation fraction of 50% using these curves. Error bars on $D_{crit}$ indicate an assumed uncertainty in DMA-transmitted particle diameter of 3.5%.

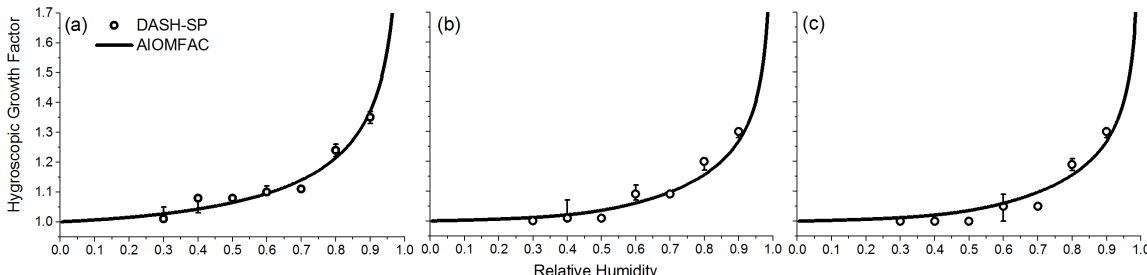

**Figure 4.** Comparison of hygroscopic growth factors measured with the DASH-SP and those predicted by AIOMFAC for (a) PEG200, (b) PEG1000, and (c) PEG10000 for particles with diameters of 250 nm. For the DASH-SP measurements, symbols indicate the average HGF and error bars indicate the maximum and minimum HGFs derived from repeat measurements.

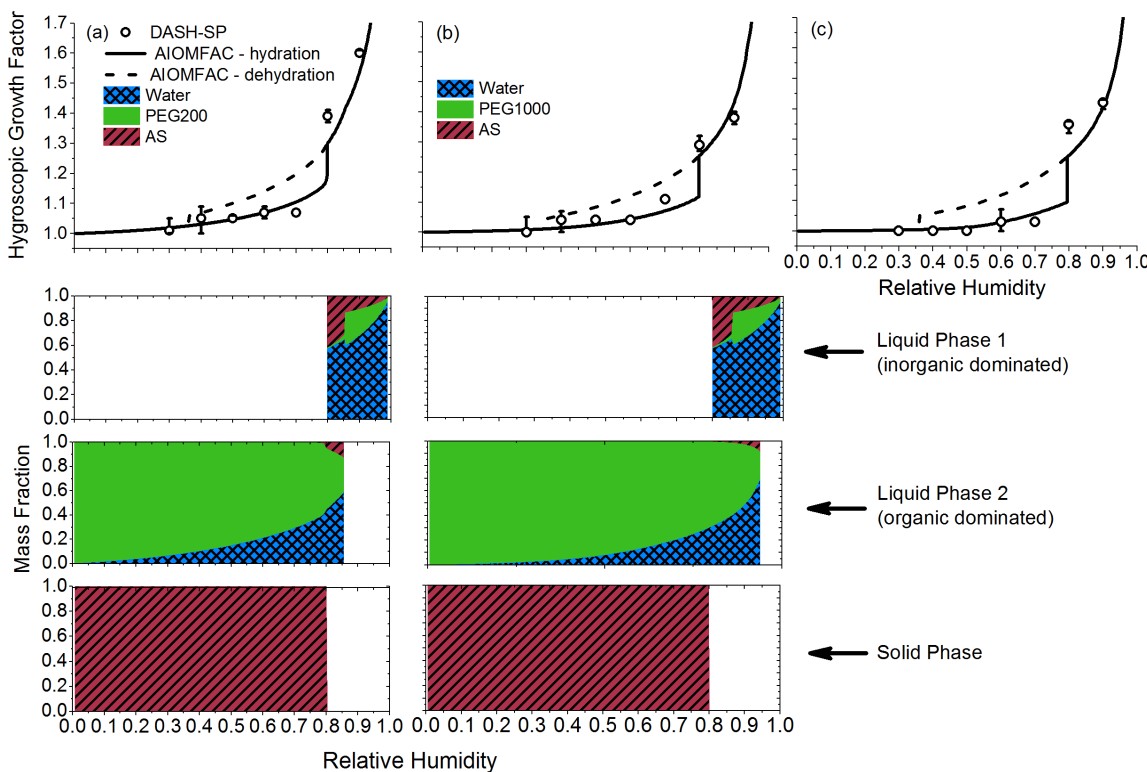

**Figure 5.** Top panels: Comparison of hygroscopic growth factors measured with the DASH-SP and those predicted by AIOMFAC for (a) PEG200-AS, (b) PEG1000-AS, and (c) PEG10000-AS for particles with diameters of 250 nm. For the DASH-SP measurements, symbols indicate the average HGF, and error bars indicate the maximum and minimum HGFs derived from repeat measurements. Panels below the growth curves show the AIOMFAC-predicted chemical composition of three potential phases present in the particles - an inorganic-dominated liquid phase, an organic-dominated liquid phase, and a solid phase - as a function of relative humidity. Further model refinements are needed before the detailed LLE phase behavior of aerosol systems containing high-molecular-mass PEG can be predicted reliably. However, all PEG-AS systems are expected to undergo liquid-liquid phase separation, with the RH at which the two separated phases merge to a single liquid phase increasing with increasing PEG molecular mass.

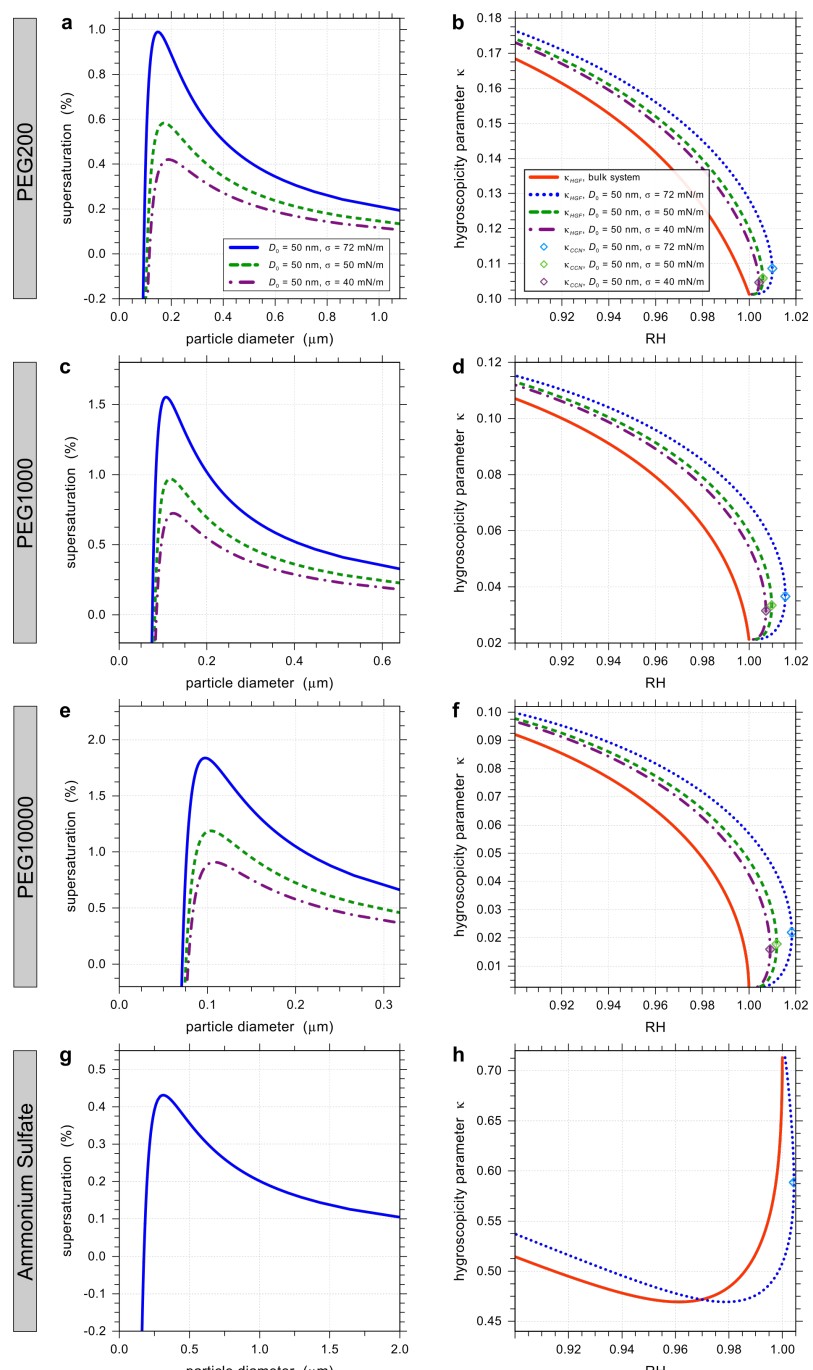

**Figure 6**. Left panels: Köhler curves (equilibrium water supersaturation vs. wet particle diameter) for the single solute PEG systems and ammonium sulfate. Right panels: predicted hygroscopicity parameter $\kappa$ in the high-RH range from 90 % RH to >100 % RH (i.e., up to supersaturated conditions with respect to liquid water). The curves are based on AIOMFAC-predicted HGF and $\kappa$-Köhler theory for particles of dry diameter $D_0 = 50$ nm with different values for the air-particle surface tension $\sigma$, as indicated in (a, b); all for a temperature of 298.15 K. The red curve shows the prediction for the bulk system for the water activity (= equilibrium RH) range from 0.9 to 0.99999. The predicted hygroscopicity parameters at CCN activation, $\kappa_{CCN}$, are shown by the open diamonds for the given particle properties ($D_0$, $\sigma$), with error bars denoting the numerical resolution of the corresponding maxima in equilibrium supersaturation (related to the maxima of the Köhler curves). Note the differences in axis scales.

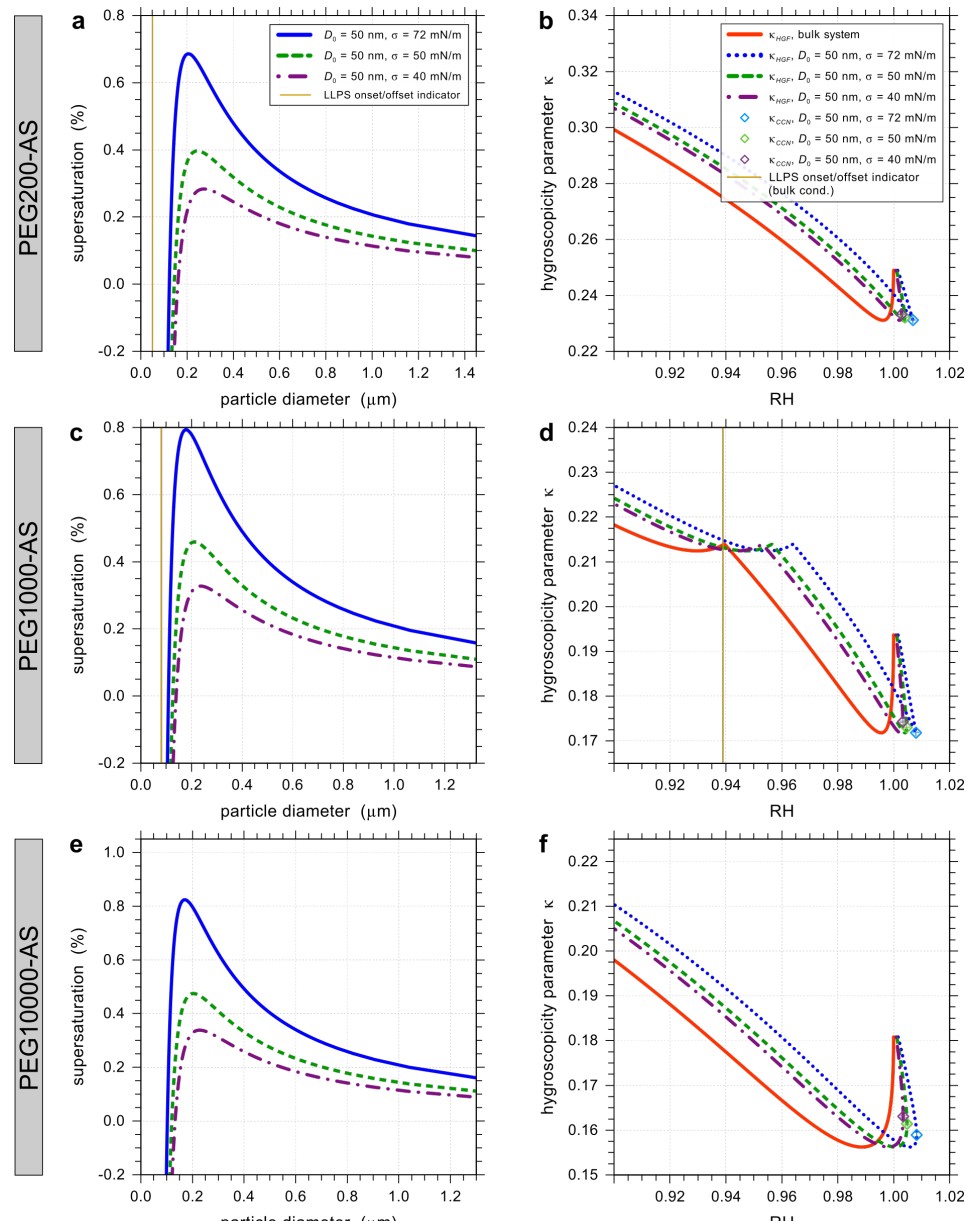

**Figure 7.** Predicted hygroscopicity properties for the mixed PEG-AS systems with a PEG:AS mass ratio of 2:1. Left panels: Köhler curves (equilibrium water supersaturation vs. wet particle diameter). Right panels: predicted hygroscopicity parameter $\kappa$ in the high-RH range from 90 % RH to >100 % RH, i.e., up to supersaturated conditions with respect to liquid water. The curves are based on AIOMFAC-predicted HGF and $\kappa$-Köhler theory for particles of dry diameter $D_0 = 50$ nm with different values for the air-particle surface tension $\sigma$, as indicated in (a, b); all for a temperature of 298.15 K. The red curve shows the prediction for the bulk system for the water activity range from 0.9 to 0.99999. The predicted hygroscopicity parameters at CCN activation, $\kappa_{CCN}$, are shown by the open diamonds for the given particle properties ($D_0$, $\sigma$), with error bars denoting the numerical resolution of the corresponding maxima in equilibrium supersaturation (related to the maxima of the Köhler curves). The yellow vertical lines indicate the onset of phase separation, with a LLPS existing at RH or particle diameter below the indicated value and a single, homogeneous liquid phase above it. The predictions for the PEG10000-AS system are here based on a ZSR approach (see Section 2.4), which treats the particle as an LLPS system for the whole RH range. Note the differences in axis scales.

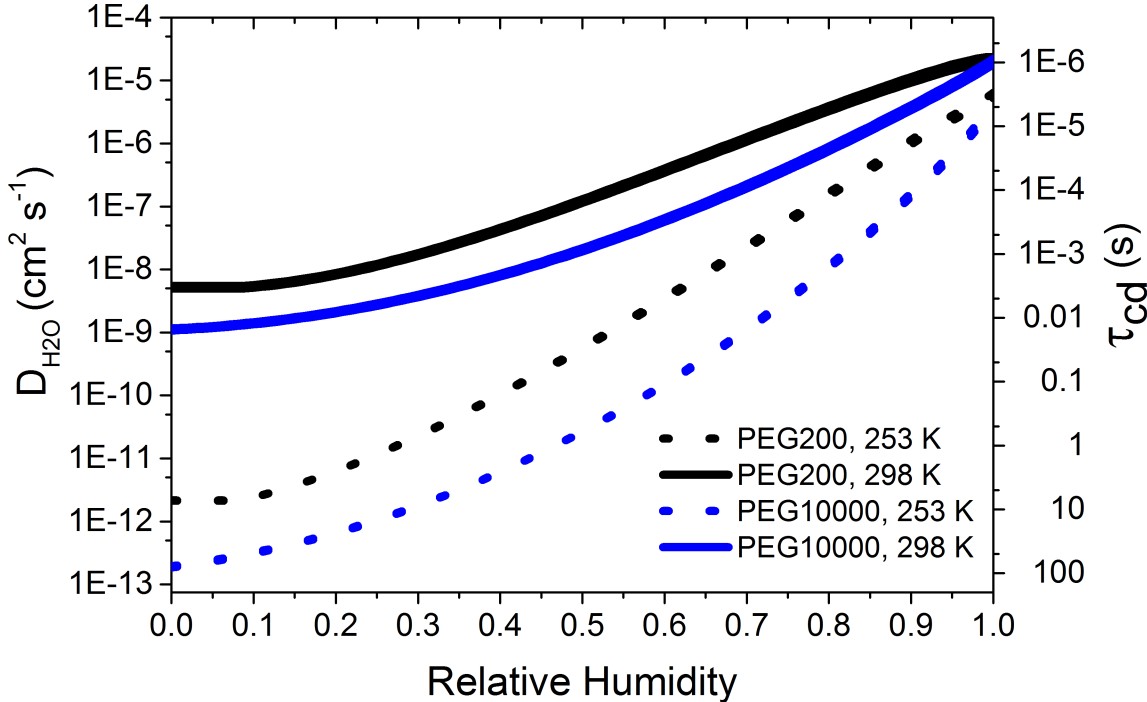

**Figure 8.** Predicted bulk diffusivity of water ($D_{H2O}$) and characteristic equilibration timescale ($\tau_{cd}$) as a function of relative humidity at 298 and 253 K. At 298 K. Water diffusion is predicted to be rapid for both PEG200 and PEG10000, and equilibration is expected to be achieved in the 4 s residence time of the DASH-SP humidifiers. At 253 K, higher particle viscosity results in equilibration timescales that approach 100 s.

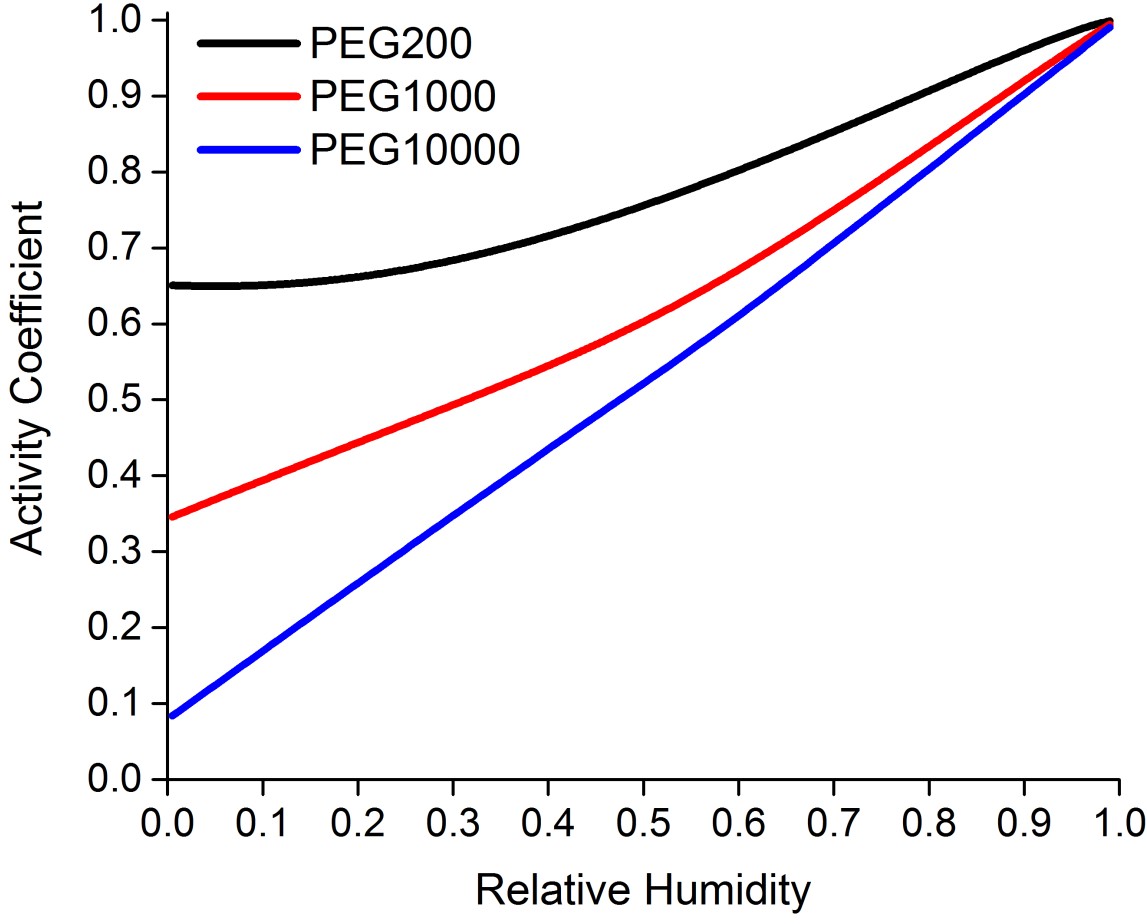

**Figure 9.** AIOMFAC predicted activity coefficients of water in PEG200, PEG1000, and PEG10000 as a function of RH. At low to moderate RH, activity coefficients are closer to unity in PEG200 as compared to the higher-molecular-mass PEG systems, indicating a lesser degree of thermodynamic non-ideality. Activity coefficients approach unity for all aerosol systems as RH approaches water saturation.

| Fit Parameter | AS | PEG200 | PEG1000 | PEG10000 | PEG200-AS | PEG1000-AS | PEG10000-AS |
|---|---|---|---|---|---|---|---|
| c1 | 4.0039 | 20.5833 | 63.4274 | 68.5808 | 3.5617 | 10.4271 | 6.5243 |
| c2 | -3.9709 | -8.7330 | -16.1516 | -27.7027 | -2.9881 | -7.0006 | -5.5021 |
| c3 | 0.6191 | -0.0930 | 0.2237 | 0.4906 | 0.5978 | 0.2788 | 0.4834 |
| c4 | 0.0294 | 0.0858 | 0.0584 | 0.0372 | 0.0285 | 0.0578 | 0.0386 |
| c5 | 1883.837 | 4429.546 | 0.5001 | 0.1481 | 1254.705 | 4039.93 | 1005.922 |

**Table A1.** Parameters describing the 5-parameter sigmoidal curves fit to measurements of CCN activation fraction. Experimental CCN activation fraction data and the best-fit sigmoidal curves are shown in Figure 2. Pearson r values exceeded 0.99 for all aerosol systems.

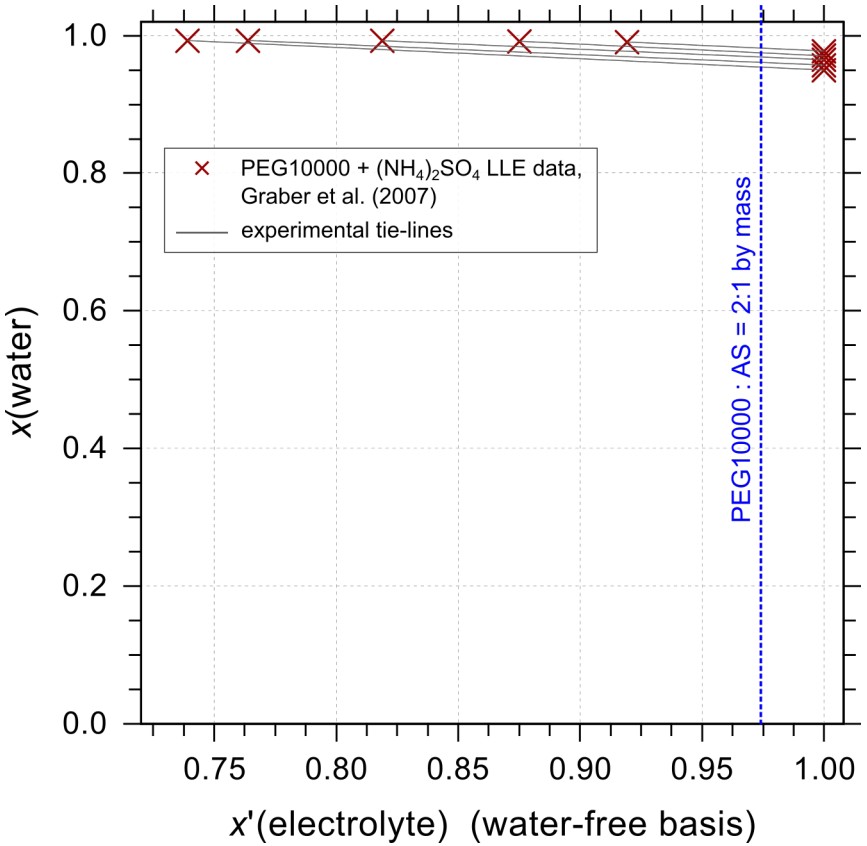

**Figure A1.** Experimental LLE tie-line data for the aqueous PEG10000-AS system at 298.15 K by Graber et al. (2007), shown as the mole fraction of water ($x$(water)) in the ternary PEG10000-AS-water liquid bulk mixture versus the mole fraction of AS ($x'$(electrolyte)) of the PEG10000-AS mixture on a water-free basis. The dashed blue line indicates the dry PEG10000:AS mass ratio of 2:1 studied in the DASH-SP experiments. Based on the observation of LLE at high mole fractions of water (relevant to high RH values), LLE is expected to persist from low RH to RH values exceeding those studied here with the DASH-SP for the PEG10000-AS aerosol system.

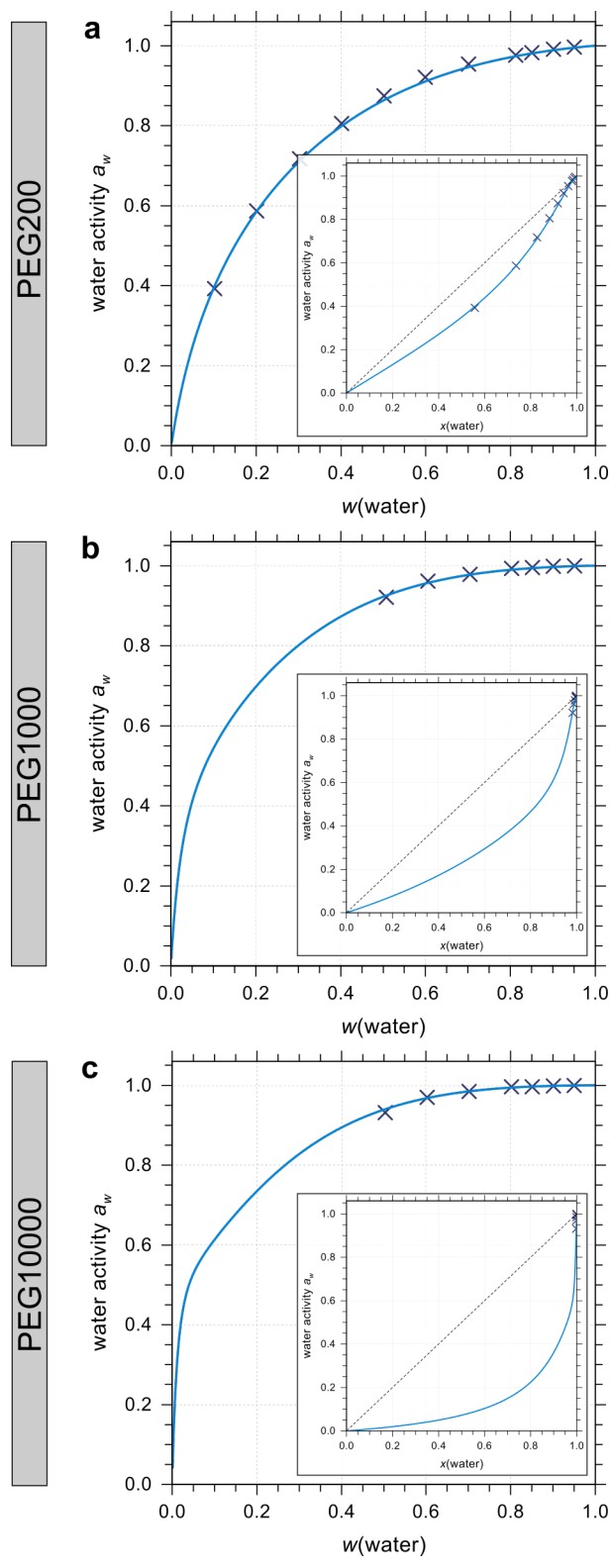

5   **Figure A2.** Comparison of measured and predicted water activity vs. mixture composition at 298.15 K. Experimental data (crosses) are from Ninni et al. (1999). The blue curves show the specific AIOMFAC

predictions for the mole fraction based water activity of aqueous PEG200 (a), PEG1000 (b) and PEG10000 (c) mixtures vs. mass fraction of water, $w$(water). The insets show water activity vs. mole fraction of water, $x$(water); the dotted diagonal line shows the water activity of a hypothetical ideal solution. The deviations between ideal and non-ideal $a_w$ curves indicate the pronounced deviation from ideal solution behavior with increasing molar mass (i.e. increasing molecular size) of the PEG component (from a to c).