# Peer review of "Discontinuities in hygroscopic growth below and above water saturation for laboratory surrogates of oligomers in organic atmospheric aerosols"

_Atmospheric Chemistry and Physics, 2016_

## Referee Comment (RC1) · Anonymous Referee #2 · 16 May 2016

Summary:

This paper presents hygroscopic growth factors (HGFs) and cloud condensation nuclei (CCN) activity of particles composed of ammonium sulfate and/or polyethylene glycol (PEG) polymers (MW = 200, 1000, or 10000 g/mol). This data is used with a water activity model to conclude that kinetic limitations to water uptake are negligible, and that differences in water uptake under sub- and supersaturated conditions are caused by "a combination of RH-dependent differences in the sensitivity of water uptake behavior to non-ideal interactions and to surface tension effects." This topic is of interest to

ACP readers, and both the data presented and the analyses are sound. I recommend publication once the comments below have been addressed.

General Comments:

In the abstract, one of the main conclusions (quoted above) regards "apparent discontinuities in hygroscopicity above and below water saturation." Two paragraphs in the introduction discuss "previous studies [that] have observed low degrees of hygroscopic growth below water saturation, but high CCN activity." Although the authors emphasize the fact that PEG10000-AS particles are more CCN active than pure AS, they do not explicitly conclude that PEG hygroscopicity is greater under supersaturated (CCN) conditions than it is under subsaturated (HGF) conditions. Instead the focus is shifted to MW dependence, and I wasn't sure whether the authors thought it was obvious that the "gap" in hygroscopicity was present, or that it was outside the scope of this work. I recommend the authors present data that would allow the difference in PEG hygroscopicity under sub- and supersaturated conditions to be evaluated. For example, CCN activity data is compared to pure AS, and this could also be done for HGFs. Or, a single hygroscopicity parameter that accounts for differences in RH and particle size could be reported for both situations.

In general the influence of water activity (the Raoult effect) on hygroscopicity is thoroughly addressed. The same cannot be said for surface tension (Kevin) explanations, however, despite the fact that this is the explanation given for a key observation in the abstract (lines 32–34): "[A]n increase in CCN activity with increasing PEG molecular mass was observed. This ... is attributed to an increase in the efficiency of PEG as a surfactant with increasing molecular mass." There are instances where the surface activity of PEG200, PEG1000, and PEG10000 are compared, but the specifics of the comparison are not clear (see Specific Comments). Also, no attempt is made to quantify these differences in surface activity. The authors could model CCN activity accounting for surface-bulk partitioning of PEG using the Szyszkowski equation or some other equation of state. Alternatively, a simple single-parameter approach could

be used. If the CCN activity of the PEG is believed to arise from its surface activity, it could be parameterized as $\delta$org, which is the thickness of the organic film at activation assuming all organic material is adsorbed to the surface (Ruehl et al., 2016). At a supersaturation of 0.6%, Dcrit of 53, 47, and 34 nm are reported for particles with a 1:2 mass ratio of ammonium sulfate to PEG200, PEG1000, and PEG10000, respectively. This is equivalent to $\delta$org of 0.19, 0.12, and 0.04 nm. For linear carboxylic diacids with carbon numbers from 3 to 8, Ruehl et al. (2016) found that $\delta$org ranged from 0.07 to 0.21 nm. The similarities in these ranges of assumed film thickness support the conclusion that PEG primarily contributes to CCN activity of mixed inorganic-organic particles through its surface activity.

The conclusion that the results reflect "RH-dependent differences in the sensitivity of water uptake behavior to non-ideal interactions" (lines 39–40 of the Abstract) should be made more clear. If the "RH-dependent differences" are between sub- and supersaturated conditions, then the conclusion is that non-ideal interactions are important at RH≤90% and negligible at RH∼100%. But doesn't this simply arise from the convention of setting the activity coefficient to unity at infinite dilution? Or, are the authors actually taking about "MW-dependent" differences? Figure 7 suggests that water activity models need to take molecular size into account, because Raoult's law will underpredict water uptake for high-MW compounds. I am not sure that the authors can conclude anything more general than this.

Specific:

Page 12, line 14 – what is meant by the "magnitude of this surface tension depression" – does this mean the minimum surface tension value possible? Or does it refer to the relationship between surface tension and concentration (e.g., a Szyszkowski parameterization)? If it has been established that PEG surface activity increases with MW, is it known that this relationship holds at the high values of water activity relevant for CCN activation? The textbook cited (Rey and May) does not provide much guidance – it might be better to cite a relevant paper from the textbook bibliography.

Page 14, line 9 – could you provide a brief explanation for why LLPS decreases water uptake? Is this always true, or just for the particle sizes and compositions in this work?

Page 14, line 29 – while I understand that the diffusivity of water in pure PEG is too fast to account for the "hygroscopicity gap", I wonder if this is relevant at all for particle hydration. Wouldn't PEG particles dissolve from the outside inward when exposed to increasing RH? And so wouldn't the diffusion of water through a (non-supersaturated) PEG aqueous solution be more relevant than through pure PEG? Note that this is different from drying a particle from the outside – in that case, a viscous supersaturated PEG "shell" would form, and any water in the droplet interior would have to diffuse through this shell before equilibrium could be reached.

Page 16, line 10 – what is meant by the "efficiency with which PEG depresses surface tension"? Is this based on the minimum value of surface tension possible? Or the magnitude of the reduction relative to PEG concentration? If the later, is this surface, bulk, or total concentration? Again, a better citation would help make this more clear.

Figure 3 – I recommend adding AIOMFAC predictions to this figure as well, or simply add predictions assuming an ideal solution. This will help put these data into context.

Citations:

Ruehl, C; Davies, J; Wilson, K: An interfacial mechanism for cloud droplet formation on organic aerosols, Science 6280, 1447-1450 (2016).

---

## Referee Comment (RC2) · Anonymous Referee #1 · 20 Jun 2016

Hodas et al. present data on hygroscopic growth and CCN activation for PEG of different molecular weights and PEG particles mixed with AS. These data are analyzed in the context of a modified version AIOMFAC for equilibrium water uptake and a viscosity/diffusion model for kinetic limitations due to diffusion. The models are used to conclude the extent to which viscosity contributes to differences in water-uptake behavior below and above water saturation that are reported in the literature.

This manuscript touches on a range of important issues that are currently debated in the literature. In general, it captures the gist of these discussions well. With appropriate revision, the manuscript is in principle publishable in Atmospheric Chemistry and Physics.

However, in it's present form, the manuscript isn't a strong contribution. The ideas revolving around viscosity are interesting and new and the assumptions made to model potential effects are quite reasonable. In contrast, the data analysis, and theoretical discussion could be significantly improved. Throughout the manuscript, various claims about agreement of the measurements with prior work and the effect of surface tension effects on CCN are not sufficiently backed up.

Major comments:

(1) The PEG water uptake measurements should be compared to bulk water activity data before using modified AIOMFAC. The data are readily available (e.g. Gaube, 1993). While AIOMFAC has it's place to interpolate activity coefficients it is more a test of how well activity coefficient models can reproduce data and less a test if the measured DASH-SP data are consistent with bulk water activity data.

(2) The CCN data analysis should be improved.

- The fits to the data in Fig. 2 are poor. For example, the black lines don't seem to go through the data at all. Even the AS control fit seems to off by a few nm, which will strongly influence the calibrated supersaturation. Issues to be addressed include: application of multiple charge correction for DMA transfer, application of loss correction in the CCN, which explains the gradual trend from 0.8 to 1 activated fraction in the plot, and application of a more appropriate fit function to better represent the data.

- Include the relationship between AS and dry diameter and sc that is assumed in the calibration. Is it possible that the calibration drifted between measurements?

- State whether or not there is an accounting for particle shape effects in the comparison with AS. How would that affect your conclusion since PEG is liquid and spherical in comparison.

[Figure]

(3) Improved CCN theoretical analysis is needed. The manuscript states that "observed increases in CCN activity with molecular mass and the enhancement in the CCN activity of PEG10000-AS compared to pure AS can likely be attributed to the fact that PEG is surface active and has been shown to lower the surface tension of the air-water interface when present in aqueous solution.". To date all experiments of mixtures of inorganic + organic surfactant have shown that the CCN activity of mixed particles is less than than what would predict even from ZSR (Prisle et al., 2010, Petters and Kreidenweis, 2013 and Petters and Petters, 2016). In other words, the data unanimously show that dissolved surfactants suppress CCN activity. These results are broadly supported by CCN theory that includes appropriate water activity and surface tension treatment in Kohler theory, although it should be understood that those descriptions still require further development. Nonetheless, the reported enhanced CCN activity of PEG10000-AS compared to pure AS is an extraordinary conclusion that requires significant elaboration. To support this finding a revised paper should include

- improved data analysis (see point 2).

- discussion of experimental issues raised in Petters and Petters (2016), which include fractionation of composition in the atomizer and serious questions regarding reproducibility of CCN measurements for surfactant aerosols. While it may well not be an issue for the particular system used here, a single CCN data point using a similar methodology is insufficient to support the claims made.

- quantitative comparison of pure PEG CCN data, including those in Petters et al. (2009) for PEG 200, 450, and 2000. Their data follow the decreasing trend one would expect with increasing molecular size from theory. It would be important to demonstrate whether or not the data agree at lower MW and how much the trend of decreasing CCN with MW reverses for the higher MWs. (Quantitative agreement between the datasets is not a necessary condition for publication; however, the reversal from a theoretically expected trend requires quantification and discussion about potential reasons for the behavior).

- quantitative surface tension data to support the stated increase in surface tension with molecular weight.

- accurate description of water activity at RH near activation, with uncertainty fits based on DASH-SP, bulk water activity data, and/or AIOMFAC.

- theoretical CCN predictions using water activity and surface tension using the available data and the authors choice of any of the widely available descriptions of CCN theory including surfactants (Sorjamaa et al., 2004, Raatikainen and Laaksonen, 2011, Topping, 2011, Petters and Kreidenweis, 2013, Ruehl et al., 2016). If satisfying closure between different measurements and measurement and theoretical predictions cannot be achieved, the conclusions in the paper should reflect those uncertainties.

"After entering the DASH-SP inlet, the aerosols are further dried in a Nafion dryer (with a residence time of 1 s), they pass through a 210 Po neutralizer, and are then size-selected with a long-column differential mobility analyzer (DMA) based on their electrical mobility"

- what was the RH? Can residual water affect the measurement?

References

Gaube, J., Pfennig, A. and Stumpf, M. 1993. Vapor-liquid-equilibrium in binary and ternary aqueous-solutions of poly(ethylene glycol) and dextran. J. Chem. Engin. Data 38, 163–166.

Sorjamaa, R., B. Svenningsson, T. Raatikainen, S. Henning, M. Bilde, and A. Laaksonen (2004), The role of surfactants in Köhler theory reconsidered, Atmos. Chem. Phys., 4(8), 2107–2117, doi:10.5194/acp-4-2107-2004.

Raatikainen, T., and A. Laaksonen (2011), A simplified treatment of surfactant effects on cloud drop activation, Geosci. Model Dev., 4(1), 107–116, doi:10.5194/gmd-4-107-2011.

[Figure]

Topping, D. 2010, An analytical solution to calculate bulk mole fractions for any number of components in aerosol droplets after considering partitioning to a surface layer, Geosci. Model Dev., 3, 635-642, doi:10.5194/gmd-3-635-2010.

Petters, M. D., S. M. Kreidenweis, A. J. Prenni, R. C. Sullivan, C. M. Carrico, K. A. Koehler, and P. J. Ziemann (2009), Role of molecular size in cloud droplet activation, Geophys. Res. Lett., 36, L22801, doi:10.1029/2009GL040131.

Petters, M. D., and S. M. Kreidenweis (2013), A single parameter representation of hygroscopic growth and cloud condensation nucleus activity—Part 3: Including surfactant partitioning, Atmos. Chem. Phys., 13(2), 1081–1091, doi:10.5194/acp-13-1081-2013.

Prisle, N. L., T. Raatikainen, A. Laaksonen, and M. Bilde (2010), Surfactants in cloud droplet activation: Mixed organic-inorganic particles, Atmos. Chem. Phys., 10(12), 5663–5683, doi:10.5194/acp-10-5663-2010.

Petters, S. S. and M. D. Petters (2016), Surfactant effect on cloud condensation nuclei for two-component internally mixed aerosols, J. Geophys. Res., 121, 1878–1895, doi:10.1002/2015JD024090.

Ruehl, C; Davies, J; Wilson, K: An interfacial mechanism for cloud droplet formation on organic aerosols, Science 6280, 1447-1450 (2016).

---

## Author Comment (AC1) · 5 Aug 2016

Discontinuities in the hygroscopic growth below and above water saturation for laboratory surrogates for oligomers in organic atmospheric aerosol

**Response to Reviewers**

We thank the reviewers for thorough and constructive reviews. Please find below our responses and revisions associated with individual comments. We feel that we have addressed each of the reviewers' comments and concerns and, in all cases, this has led to improvements in the manuscript. Please note that responses to reviewer 2 are provided first, as this reflects the chronological order in which comments were received and addressed. In many cases, responses to reviewer 1 follow from revisions made in our initial response to reviewer 2's comments.

**Anonymous Referee #2 Received and published: 16 May 2016**

**Summary:**

This paper presents hygroscopic growth factors (HGFs) and cloud condensation nuclei (CCN) activity of particles composed of ammonium sulfate and/or polyethylene glycol (PEG) polymers (MW = 200, 1000, or 10000 g/mol). This data is used with a water activity model to conclude that kinetic limitations to water uptake are negligible, and that differences in water uptake under sub- and supersaturated conditions are caused by "a combination of RH-dependent differences in the sensitivity of water uptake behavior to non-ideal interactions and to surface tension effects." This topic is of interest to ACP readers, and both the data presented and the analyses are sound. I recommend publication once the comments below have been addressed.

**General Comments:**

1. In the abstract, one of the main conclusions (quoted above) regards "apparent discontinuities in hygroscopicity above and below water saturation." Two paragraphs in the introduction discuss "previous studies [that] have observed low degrees of hygroscopic growth below water saturation, but high CCN activity." Although the authors emphasize the fact that PEG10000-AS particles are more CCN active than pure AS, they do not explicitly conclude that PEG hygroscopicity is greater under supersaturated (CCN) conditions than it is under subsaturated (HGF) conditions. Instead the focus is shifted to MW dependence, and I wasn't sure whether the authors thought it was obvious that the "gap" in hygroscopicity was present, or that it was outside the scope of this work. I recommend the authors present data that would allow the difference in PEG hygroscopicity under sub- and supersaturated conditions to be evaluated. For example, CCN activity data is compared to pure AS, and this could also be done for HGFs. Or, a single hygroscopicity parameter that accounts for differences in RH and particle size could be reported for both situations.

Reviewer 2 raises a good point. In the initial paper version we focus on the discontinuity in the influence of PEG molecular weight/viscosity on water uptake below and above water saturation, but do not explicitly discuss differences in hygroscopicity above and below water saturation for the individual PEG components. We appreciate the suggestions provided by reviewer 2 and have made several revisions to the manuscript in response. First, in the revised manuscript, HGFs for the aerosol systems are compared to those for pure AS. We have also calculated values of kappa

from measurements of HGFs at 90%, as well as based on measured CCN data and have added text exploring differences in apparent hygroscopicity above and below water saturation. Finally, a thorough exploration of variability in kappa as a function of RH was performed using the AIOMFAC-based equilibrium model. Specifically, kappa values were calculated for RH > 90% for 50 nm particles, assuming a range of droplet surface tensions. A summary of the revisions made to the text is below:

[revised manuscript text omitted]

2. In general the influence of water activity (the Raoult effect) on hygroscopicity is thoroughly addressed. The same cannot be said for surface tension (Kevin) explanations, however, despite the fact that this is the explanation given for a key observation in the abstract (lines 32-34): "[A]n increase in CCN activity with increasing PEG molecular mass was observed. This...is attributed to an increase in the efficiency of PEG as a surfactant with increasing molecular mass." There are instances where the surface activity of PEG200, PEG1000, and PEG10000 are compared, but the specifics of the comparison are not clear (see Specific Comments). Also, no attempt is made to quantify these differences in surface activity. The authors could model CCN activity accounting for surface-bulk partitioning of PEG using the Szyszkowski equation or some other equation of state. Alternatively, a simple single-parameter approach could be used. If the CCN activity of the PEG is believed to arise from its surface activity, it could be parameterized as  $\delta org$ , which is the thickness of the organic film at activation assuming all organic material is adsorbed to the surface (Ruehl et al., 2016). At a supersaturation of 0.6%, Dcrit of 53, 47, and 34 nm are reported for particles with a 1:2 mass ratio of ammonium sulfate to PEG200, PEG1000, and PEG10000, respectively. This is equivalent to dorg of 0.19, 0.12, and 0.04 nm. For linear carboxylic diacids with carbon numbers from 3 to 8, Ruehl et al. (2016) found that  $\delta$  org ranged from 0.07 to 0.21 nm. The similarities in these ranges of assumed film thickness support the conclusion that PEG primarily contributes to CCN activity of mixed inorganicorganic particles through its surface activity.

We agree with the reviewer that additional analyses and discussion of the influence of surface tension is a valuable addition to the manuscript, and that the calculation of  $\delta$  org provides a valuable approach for evaluating differences in surface activity across the aerosol systems. We also appreciate the detail provided by the reviewer regarding previous calculations of  $\delta$  org in Ruehl et al. (2016). In addition to adding a discussion of this to the manuscript, we have also calculated Kohler curves for 50 nm particles and made calculations of  $D_{crit}$  at a supersaturation of 0.8% using Köhler theory, assuming a range of droplet surface tensions. Calculated values of  $D_{crit}$  were then compared to values obtained from CCN measurements to determine the range of surface tensions that led to best agreement between and theory and experimental results. We have updated the text to reflect these calculations and to add a discussion of their implications, as summarized below. Please also see text from the comment above describing relevant changes to the text.

[revised manuscript text omitted]

3. The conclusion that the results reflect "RH-dependent differences in the sensitivity of water uptake behavior to non-ideal interactions" (lines 39–40 of the Abstract) should be made more clear. If the "RH-dependent differences" are between sub- and supersaturated conditions, then the conclusion is that non-ideal interactions are important at RH\_90% and negligible at RH\_100%. But doesn't this simply arise from the convention of setting the activity coefficient to unity at infinite dilution? Or, are the authors actually taking about "MW-dependent" differences? Figure 7 suggests that water activity models need to take molecular size into account, because Raoult's law will underpredict water uptake for high-MW compounds. I am not sure that the authors can conclude anything more general than this.

This statement was meant largely to reflect that our observations support the finding from Wex et al. (2008) that at the more concentrated conditions relevant to hygroscopic growth below water saturation, variables influencing water activity dominate in driving variability in water uptake. While at higher RH and conditions relevant to CCN activity, the influence of surface tension becomes important. Figure 8 illustrates the modeled RH-dependent differences in water activity. Please note that the activity coefficients for organics and water are not set to unity at infinite dilution. This is only the case for dissolved electrolytes, and this thermodynamic reference state is properly accounted for by the AIOMFAC model. The activity coefficients of the PEGs at dilute conditions in water (near 100 % RH) can in fact be very different from unity due to their large difference in molecular shape and size in comparison to water. As shown below, we have reworded the statement to reflect the focus of the paper on the influence of surface tension. As suggested by the reviewer, we have also added text noting differences in the degree of under

production of water uptake by Raoult's law with molecular mass:

"A comparison of experimental and modeling investigations of water-uptake behavior of PEGs with a range of molecular masses and viscosities suggests that such discontinuities in apparent hygroscopicity above and below water saturation can be attributed, at least in part, to differences in the sensitivity of water uptake behavior to surface tension effects."

"It is evident that activity coefficients are substantially lower for the higher molecular mass PEGs up to an  $RH \sim 95\%$ , indicating a greater degree of non-ideality for those solutions. This suggests that the degree to which Raoult's law will under predict water uptake is greater for high molecular mass compounds, indicating the importance of accounting for molecular size in water activity models."

Specific:

4. Page 12, line 14 – what is meant by the "magnitude of this surface tension depression" – does this mean the minimum surface tension value possible? Or does it refer to the relationship between surface tension and concentration (e.g., a Szyszkowski parameterization)? If it has been established that PEG surface activity increases with MW, is it known that this relationship holds at the high values of water activity relevant for CCN activation? The textbook cited (Rey and May) does not provide much guidance – it might be better to cite a relevant paper from the textbook bibliography.

We have re-worded this statement to reflect our meaning of differences in droplet surface tension across the PEG oligomers. Please also see the response to comment 2 above regarding a quantitative exploration of these differences in droplet surface tension across the aerosol systems. These calculations suggest that the surface tensions near droplet activation is clearly influenced by the presence of PEG oligomers and tends to be significantly lower than the surface tension of pure water. As suggested by the reviewer, we have also updated references to include more specific papers from the textbook initially cited.

Re-worded sentence: "There is evidence for decreases in surface tension with increasing PEG molecular mass (Rey and May, 2010; Winterhalter et al., 1995)."

**5. Page 14, line 9 – could you provide a brief explanation for why LLPS decreases water uptake? Is this always true, or just for the particle sizes and compositions in this work?**

Recent work has explored the influence of LLPS on water uptake. For example, Hodas et al. (2015) ACP illustrated that the assumption of an ideal mixture generally led to over prediction of water uptake, particularly at low to moderate RH, than models that accounted for non-ideal interactions. The extent to which a ZSR-like calculation well-predicted water uptake behavior depended on the prevalence of LLPS and the RH range over which LLPS persisted. Renbaum-Wolff et al. (2016) illustrated that LLPs at high RH was associated with enhanced CCN activity. For the present manuscript, The LLPS state with partial miscibility leads to more water uptake than the ZSR-like calculation with complete separation of organic and inorganic species. This is an effect of having two phases that both contain some amount of PEG and AS, which leads to a higher water uptake into each of these phases near the LLPS onset/merging of the two phases to one liquid phase. Water can be considered the common solvent for the dissolved ions and PEG molecules, which moderates unfavorable interactions between the organic and the ions. This feature of mixture water content has been predicted for many other systems and is explained in

Zuend et al. (2010).

6. Page 14, line 29 – while I understand that the diffusivity of water in pure PEG is too fast to account for the "hygroscopicity gap", I wonder if this is relevant at all for particle hydration. Wouldn't PEG particles dissolve from the outside inward when exposed to increasing RH? And so wouldn't the diffusion of water through a (non-supersaturated) PEG aqueous solution be more relevant than through pure PEG? Note that this is different from drying a particle from the outside – in that case, a viscous supersaturated PEG "shell" would form, and any water in the droplet interior would have to diffuse through this shell before equilibrium could be reached.

We agree with the reviewer that model results are most relevant to the potential circumstance discussed in the manuscript in which the formation of a viscous "shell" during the particle drying process inhibits the evaporation of water ("*if particles did not achieve equilibrium with water vapor in the*  $\sim$ 5 *s residence time of the DASH-SP and diffusion dryers and/or in the 4 s residence time of the DASH-SP humidifier, it would be expected that experimental observations would deviate substantially from growth curves predicted by AIOMFAC*"). Modeled water diffusion in pure PEG will likely be a lower limit for diffusivity, as water is diffusing partially through an aqueous PEG solution of lower viscosity as a particle gets exposed to elevated RH and takes up water from the outside toward the center. The reviewer raises a good point that this should be clarified in the manuscript. We have added the following text to address this:

"During particle hydration, the diffusion of water through a non-supersaturated aqueous PEG solution would be more representative, due to the partial dissolution of the PEG-containing aerosols with increasing RH. Modeled conditions are more relevant to diffusion through a PEG shell, which may form during the rapid drying of the particles studied here. Thus, modeled water diffusion in pure PEG will likely be a lower limit for diffusivity."

7. Page 16, line 10 – what is meant by the "efficiency with which PEG depresses surface tension"? Is this based on the minimum value of surface tension possible? Or the magnitude of the reduction relative to PEG concentration? If the later, is this surface, bulk, or total concentration? Again, a better citation would help make this more clear.

We have rephrased this statement to reflect that the effectiveness with which the PEG oligomers depress droplet surface tension depends is related to enhanced bulk-to-surface partitioning with increasing polymer size of the PEGs. The decrease in surface tension with increase in polymer size (and molar mass) could further support the idea that both bulk-to-surface partitioning may be enhanced for the larger PEGs with lower pure component surface tension values, as indicated by our comparison in (the new) Fig. 3. The sentence was rephrased to read:

"Because the effectiveness with which the PEG oligomers depress the surface tension of the airparticle interface is expected to increase with the molecular mass of the PEG polymer (Rey and May, 2010; Winterhalter et al., 1995), likely as a result of enhanced bulk-to-surface partitioning with increasing polymer size..."

8. Figure 3 – I recommend adding AIOMFAC predictions to this figure as well, or simply add predictions assuming an ideal solution. This will help put these data into context.

Please see our response to comment 2.

Anonymous Referee #1 Received and published: 20 June 2016

Hodas et al. present data on hygroscopic growth and CCN activation for PEG of different molecular weights and PEG particles mixed with AS. These data are analyzed in the context of a modified version AIOMFAC for equilibrium water uptake and a viscosity/diffusion model for kinetic limitations due to diffusion. The models are used to conclude the extent to which viscosity contributes to differences in water-uptake behavior below and above water saturation that are reported in the literature.

This manuscript touches on a range of important issues that are currently debated in the literature. In general, it captures the gist of these discussions well. With appropriate revision, the manuscript is in principle publishable in Atmospheric Chemistry and Physics.

However, in it's present form, the manuscript isn't a strong contribution. The ideas revolving around viscosity are interesting and new and the assumptions made to model potential effects are quite reasonable. In contrast, the data analysis, and theoretical discussion could be significantly improved. Throughout the manuscript, various claims about agreement of the measurements with prior work and the effect of surface tension effects on CCN are not sufficiently backed up.

Please see responses to individual comments below.

Major comments:

(1) The PEG water uptake measurements should be compared to bulk water activity data before using modified AIOMFAC. The data are readily available (e.g. Gaube, 1993). While AIOMFAC has it's place to interpolate activity coefficients it is more a test of how well activity coefficient models can reproduce data and less a test if the measured DASH-SP data are consistent with bulk water activity data.

We have added a figure and text to the appendix comparing measured and predicted water activity versus mixture composition at 298.15 K. We thank the reviewer for providing a potential reference for this effort; however, the data from Gaube (1993) are relevant only for dilute (high water activity) conditions. We have compared modeled data against results from Ninni et al. (1999), which cover a ride range of conditions, particularly for PEG200. We observed excellent agreement between AIOMFAC-predicted and measured water uptake of the binary system. Thus, the comparisons between DASH-SP measurements and AIOMFAC predictions of growth factors are sufficiently representative of a comparison between DASH-SP measurements and water-uptake for bulk conditions. The new figure, figure caption, and text added to the appendix are provided below:

In the main text: "AIOMFAC predictions of water uptake for the PEG systems are also in excellent agreement with experimental bulk water activity data for these systems (Appendix A, Fig. A2), suggesting that agreement between DASH-SP measurements and AIOMFAC-based

predictions also indicate that measured HGFs are consistent with bulk water activity data measurements."

**Figure A2.** Comparison of measured and predicted water activity vs. mixture composition at 298.15 K. Experimental data (crosses) are from Ninni et al. (1999). The blue curves show the specific AIOMFAC predictions for the mole fraction based water activity of aqueous PEG200 (a), PEG1000 (b) and PEG10,000 (c) mixtures vs. mass fraction of water, w(water). The insets show water activity vs. mole fraction of water, x(water); the dotted diagonal line shows the water activity of a hypothetical ideal solution. The deviations between ideal and non-ideal  $a_w$  curves indicate the pronounced deviation from ideal solution behavior with increasing molar mass (i.e. increasing molecular size) of the PEG component (from a to c).

"Appendix A: AIOMFAC predictions of water uptake were compared to previously published bulk water activity data for the three PEG oligomers studied here (Fig A2). Excellent agreement between AIOMFAC-predicted and measured (Ninni et al., 1999) bulk water activity data indicates that the AIOMFAC model represents the bulk diameter growth factor curves well (assuming no exotic molar volume (density) excess effects occur). Particle diameter growth factors discussed in Section 2.4 were determined from the calculated particle mass at a certain RH with the use of pure-component densities and assumption of linear additivity of the calculated component volumes. The good agreement between AIOMFAC-predicted and measured water uptake for bulk conditions indicates that comparisons between DASH-SP measurements and AIOMFAC predictions of particle diameter growth factors, discussed in Section 3.2 are essentially equivalent to a comparison between DASH-SP measurements and measured water-uptake of bulk solutions. Figure A2 also indicates that assuming activity coefficients of unity with component mole fractions representing mixture composition would result in substantial error, as the non-ideality of aqueous PEG mixtures is large due to the pronounced difference in molecular size/mass with increasing PEG polymer chain-length compared to the size of water molecules."

**(2) The CCN data analysis should be improved.**

- The fits to the data in Fig. 2 are poor. For example, the black lines don't seem to go through the data at all. Even the AS control fit seems to off by a few nm, which will strongly influence the calibrated supersaturation. Issues to be addressed include: application of multiple charge correction for DMA transfer, application of loss correction in the CCN, which explains the gradual trend from 0.8 to 1 activated fraction in the plot, and application of a more appropriate fit function to better represent the data.

We thank the reviewer for brining the issue of particle loss correction in the CCN instrument to our attention. Loss corrections were made in the revised manuscript based on the results of Brechtel and Kreidenweiss (1999), who provided a polynomial curve describing particle losses in CCN instruments as a function of particle size. In addition, to ensure a good fit between the experimental data and the fit function, we now utilize a 5-parameter sigmoidal fit function (shown below) to fit the experimental data.  $D_{crit}$  was then calculated from this curve as the diameter associated with an activation fraction of 50%. Comparison between the fitted curves and experimental data indicate good fits for all systems, as characterized by Pearson r values exceeding 0.99. The combination of the new curve fits and the particle loss correction for both experimental and calibration data indicated that the supersaturation within the instrument was higher than initially calculated in the initial calibration experiments. This has been updated in the manuscript.

"Correction factors were applied to measured CCN concentrations to account for particle losses in the CCN counter based on the results of Brechtel and Kreidenweis (1999), who provided a polynomial curve describing particle losses as a function of particle size. Loss-corrected measured activation fractions as a function of particle diameter were fit with a 5-parameter sigmoidal curve."

The decision not to apply a multiple charging correction was based on the results of Petters et al. (2007) who showed that such factors are not expected to contribute substantially to error in  $D_{crit}$  for the conditions under which experiments in this work were carried out. That work showed that the effects of multiple charging are diminished for the smaller particle sizes activated at supersaturations > 0.3%. The supersaturation studied in this work falls well above that threshold. In addition, under laboratory conditions in which an atomizer is used to generate aerosol particles, the size distribution tends to fall in a relatively narrow range, with the majority of particles having diameters smaller than those requiring a correction for multiple charging. We have added the following text to the manuscript to clarify this:

"For the experimental conditions under which CCN measurements were made in this work, the effects of multiply charged particles are not expected to be a substantial contributor to experimental uncertainty. Specifically, such effects are diminished when CCN counter supersaturations exceed 0.3% and when aerosol particles are generated through the atomization of aqueous solutions (Petters et. al., 2007).

- Include the relationship between AS and dry diameter and Sc that is assumed in the calibration. Is it possible that the calibration drifted between measurements?

All experiments were run under the same delta T and flow conditions, with the goal of exploring relative differences in CCN activity between the aerosol systems. Instrument operation was normal during experimental runs and, as a result, there is no reason to believe there were drifts in supersaturation calibration. Along the same lines, CCN measurements were conducted in close succession, with only the amount of time needed to ensure total clean out of the atomizer, sampling lines, and instrumentation waited between runs for each aerosol system. Finally, because the CCN properties of AS are well-characterized and have been shown to be well described by Kohler theory, CCN measurements for the AS control serve not only as a well-characterized metric against which the PEG-containing systems can be compared, but also serves as a marker for instrument operation by serving as snapshot of supersaturation during experiments. The critical diameter and kappa values determined from CCN measurements are also in agreement with theory and previous experiments, indicating reasonable results and proper instrument operation of Kohler theory described in Gysel et al. (2002) and verified against calculations made with the ADDEM model (Topping et al., 2005).

**- State whether or not there is an accounting for particle shape effects in the comparison with AS. How would that affect your conclusion since PEG is liquid and spherical in comparison.**

Due to uncertainty in particle shape correction factors, we assumed all particles were spherical, regardless of composition. Crystallization of AS and the potential for PEG1000 and PEG10000 to be solid after the drying step prior to size-selection in the DMA could lead to a small degree of uncertainty in dry particle diameter; however, it is expected that the influence of particle shape factor would be minimal and would not have a significant impact on our conclusions. For example, it is expected that this would be on the order of 0 - 4% for AS, based on estimated shape factors ranging from 1.0 - 1.4 (Gysel et al., 2002; Zelenyuk et al., 2006). We have added text to note the assumption of spherical particles, as well as the potential uncertainty related to the issue of the transmission of non-spherical particle through the DMA.

"All particles were assumed to be spherical. Due to the particle drying step prior to size selection in the DMA, the crystallization of AS, as well as the potential for the higher molecular mass PEG oligomers to be present as solids, could result in a small degree of uncertainty in the actual sphere-equivalent diameter of particles transmitted through the DMA. Uncertainty in particle diameter is expected to be minimal, however, as shape factors for AS have been estimated to be between 1.0 (i.e., spherical) and 1.04 (Gysel et al., 2002; Zelenyuk et al., 2006). Shape factors for PEG-containing submicron particles are unknown at present."

(3) Improved CCN theoretical analysis is needed. The manuscript states that "observed increases in CCN activity with molecular mass and the enhancement in the CCN activity of PEG10000-AS compared to pure AS can likely be attributed to the fact that PEG is surface active and has been shown to lower the surface tension of the air-water interface when present in aqueous solution." To date all experiments of mixtures of in- organic + organic surfactant have shown that the CCN activity of

mixed particles is less than what would predict even from ZSR (Prisle et al., 2010, Petters and Kreidenweis, 2013 and Petters and Petters, 2016). In other words, the data unanimously show that dissolved surfactants suppress CCN activity. These results are broadly supported by CCN theory that includes appropriate water activity and surface tension treatment in Kohler theory, although it should be understood that those descriptions still require further development. Nonetheless, the reported enhanced CCN activity of PEG10000-AS compared to pure AS is an extraordinary conclusion that requires significant elaboration. To support this finding a revised paper should include

- improved data analysis (see point 2).
- Please see responses to comment 2.

- discussion of experimental issues raised in Petters and Petters (2016), which include fractionation of composition in the atomizer and serious questions regarding reproducibility of CCN measurements for surfactant aerosols. While it may well not be an issue for the particular system used here, a single CCN data point using a similar methodology is insufficient to support the claims made.

We agree that it is important to consider limitations and thank the reviewer for bringing this publication to our attention. We have added in a discussion of potential limitations in experimental methods. In support of our conclusions, it is important to note that a trend in CCN activity is observed across the systems (i.e., increasing CCN activity with increasing PEG molecular mass) and conclusions are not based solely on a single CCN data point for a single aerosol system. Further, we have utilized several tools, including theoretical calculations with the AIOMFAC model, to explore the hygroscopic properties of the aerosol systems studied and conclusions are based on the aggregation of results from all exploratory analyses. Please also note several recent studies referenced in the manuscript indicating an influence of surface tension depression by organic components on enhanced CCN activity. Additional text regarding potential limitations has been added to the revised manuacript:

"Previous work has suggested that incomplete mixing of aerosol components in aqueous solution and/or fractionation of components during atomization can contribute to variability and uncertainty in hygroscopicity measurements for aerosol systems containing surface-active components, particularly for components with low water solubility (Petters and Petters, 2016). However, because PEG is highly water soluble, it is not expected that this was a significant contributor to uncertainty in experimental results."

- quantitative comparison of pure PEG CCN data, including those in Petters et al. (2009) for PEG 200, 450, and 2000. Their data follow the decreasing trend one would expect with increasing molecular size from theory. It would be important to demonstrate whether or not the data agree at lower MW and how much the trend of decreasing CCN with MW reverses for the higher MWs. (Quantitative agreement between the datasets is not a necessary condition for publication; however, the reversal from a theoretically expected trend requires quantification and discussion about potential reasons for the behavior).

We have added comparisons of water uptake for PEG-containing systems with those obtained in previous work. Contributors to agreement and disagreement are also discussed:

"HGFs measured here are within  $\sim$ 5% of those measured for 100 nm particles comprised of PEG with average molecular masses of 600 and 3,400 measured by Petters et al., (2006)."

"For PEG200, the value of  $\kappa_{CCN,app}$  derived here from CCN measurements falls well within the range previously reported for particles comprised of Tetraethylene glycol and Pentaethylene glycol (molecular masses = 194 and 238 g/mol, respectively;  $\kappa_{CCN,app} = 0.057 - 0.195$ ) (Petters et al., 2009b). For PEG1000, the  $\kappa_{CCN,app}$  observed here is higher than the upper limit of the range reported by Petters et al., (2009b) (0.033 – 0.064), but lower than the upper limit reported in that study for PEG400 (0.05 – 0.106). Potential contributors to differences in estimates of  $\kappa_{CCN,app}$  for PEG1000 include differences in experimental conditions for the CCN measurements (e.g., particle size and/or supersaturation), differences in the proportions of PEG oligomers with varying chain lengths resulting in an average molecular mass of 1000 g/mol, as well as measurement uncertainty."

"AIOMFAC-predicted values of  $\kappa_{CCN}$  for the PEG200 and PEG1000 aerosol systems (0.108 and 0.037, respectively for an assumed surface tension of 72 mN/m) are in very good agreement with the range of values previously reported for Tetra- and Pentaethylene glycol (comparable to PEG200) and for PEG1000 (Petters et al., 2009b)."

**- quantitative surface tension data to support the stated increase in surface tension with molecular weight.**

Please see the response to reviewer 2's comment 2 above describing the addition of a various approaches to provide a quantitative assessment of differences in surface tension across the PEG-containing aerosol systems.

**- accurate description of water activity at RH near activation, with uncertainty fits based on DASH-SP, bulk water activity data, and/or AIOMFAC.**

We have added calculations of hygroscopicity properties of the aerosol systems at RH values approaching 100% ("To further explore contributors to differences in water-uptake behavior across aerosol systems, as well as differences in apparent hygroscopicity under suband supersaturated conditions within individual aerosol systems, we performed AIOMFAC calculations with a high resolution for the high-RH range above 90 % RH toward 100 % RH with respect to bulk systems..."

We have also added substantial discussion regarding the implications of calculations of water activity and phase behavior of the systems studied here based on these calculations (Please see the newly added section 3.3 - also summarized in the response to Reviewer 2's comment 1).

- theoretical CCN predictions using water activity and surface tension using the available data and the authors choice of any of the widely available descriptions of CCN theory including surfactants (Sorjamaa et al., 2004, Raatikainen and Laaksonen, 2011, Topping, 2011, Petters and Kreidenweis, 2013, Ruehl et al., 2016). If satisfying closure between different measurements and measurement and theoretical predictions cannot be achieved, the conclusions in the paper should reflect those uncertainties.

Please see the response to reviewer 2's comment 2 above describing the addition of a quantitative assessment of differences in surface tension across the PEG-containing aerosol systems. Agreement between measurements and theoretical predictions was achieved when using a

reasonable range of surface tensions that agree with previous measurements of the surface tension of PEG.

"After entering the DASH-SP inlet, the aerosols are further dried in a Nafion dryer (with a residence time of 1 s), they pass through a 210 Po neutralizer, and are then size-selected with a long-column differential mobility analyzer (DMA) based on their electrical mobility"

**what was the RH? Can residual water affect the measurement?**

As noted in the manuscript, the particles were dried first in a diffusion dryer with a residence time of approximately 5 s and then Nafion dryer in the DASH-SP system. It is expected that RH is near zero and measurements conducted in the dry channel of the DASH-SP are less than 10%. Our calculations of diffusivity of water in PEG suggest that drying time is sufficient; however, we do include a discussion of the potential impacts of residual water (e.g, due to the formation of viscous shell during particle drying) in several places throughout the manuscript:

[revised manuscript text omitted]

activated to form CCN at a constant supersaturation (i.e., here at 0.8%). Prior to entering the inlet of the CCN counter, particles were size-selected with a long-column DMA and CCN counts were obtained for particles with dry diameters between 20 and 210 nm with a spacing of 10 nm. All particles were assumed to be spherical. Due to the particle drying step prior to size selection in the DMA, the

- 5 crystallization of AS, as well as the potential for the higher molecular mass PEG oligomers to be present as solids, could result in a small degree of uncertainty in the actual sphere-equivalent diameter of particles transmitted through the DMA. Uncertainty in particle diameter is expected to be minimal, however, as shape factors for AS have been estimated to be between 1.0 (i.e., spherical) and 1.04 (Gysel et al., 2002; Zelenyuk et al., 2006). Shape factors for PEG-containing
- 10 submicron particles are unknown at present. A condensation particle counter (CPC) sampled particles in parallel with the CCN counter to provide total particle counts, and activation fractions were calculated from the ratio of CCN concentration ( $C_{CCN}$ ) to total particle concentration measured with the CPC ( $C_{CPC}$ ). Correction factors were applied to measured CCN concentrations to account for particle losses in the CCN counter based on the results of Brechtel and Kreidenweis (2000), who provided a polynomial curve describing particle losses as a function of particle size. Loss-corrected measured activation fractions as a

function of particle diameter were fit with a 5-parameter sigmoidal curve with the following form;

15

 $C_{CCN}/C_{CPC} = MIN \left\{ \frac{c_4 \times \sqrt{D_0} + c_3}{\left[1 + \left(\frac{D_0}{c_4}\right)^{c_2}\right]^{c_5}}, 1.0 \right\}$

where c1, c2, c3, c4, and c5 are fit parameters (Table A1) and Do is the dry particle diameter. Dcrit, 
[revised manuscript text omitted]